# Hippocampal OLM interneurons regulate CA1 place cell plasticity and remapping

Matt Udakis, Matthew D. B. Claydon ⬮, Heng Wei Zhu ⬮, Elsa C. Oakes ⬮ & Jack R. Mellor ⬮ ✉

OLM interneurons selectively target inhibition to the distal dendrites of CA1 pyramidal cells in the hippocampus but the role of this unique morphology in controlling place cell physiology remains a mystery. Here we show in mice that OLM activity prevents associative synaptic plasticity at Schaffer collateral synapses on CA1 pyramidal cells by inhibiting dendritic $Ca^{2+}$ signalling initiated by entorhinal synaptic inputs. Furthermore, we find that OLM activity is reduced in novel environments suggesting that reducing OLM activity and thereby enhancing excitatory synaptic plasticity is important for the formation of new place cell representations. Supporting this, we show that selectively increasing OLM activity in novel environments enhances place cell stability and reduces remapping of newly formed place cells whilst increasing OLM activity in familiar environments led only to a transient silencing of place cells. Our results therefore demonstrate a critical role for distal dendrite targeting interneurons in regulating plasticity of neuronal representations.

The hippocampus enables us to learn and adapt to new situations by maintaining a dynamic representation of the environment in the activity of place cells. This location specific place cell activity allows quantifiable measurement of the neuronal representation of an environment and when a new environment is encountered, hippocampal place cells rapidly adapt their firing (or remap) to efficiently encode the new features of the environment[1–4]. This dynamic neuronal representation is essential for behavioural adaptation to new environments[5]. Synaptic plasticity triggered by $Ca^{2+}$ influx to dendrites and spines is a fundamental process that underpins adaptation of place cell firing, where large dendritic $Ca^{2+}$ events termed plateau potentials have been shown to be particularly effective at driving the formation and tuning of place cells[6–13]. In CA1 pyramidal cells, plateau potentials are thought to be driven by supra-linear integration of coincident CA3 (proximal) and entorhinal (distal) inputs, but the regulation of this integration, and therefore of behaviourally relevant synaptic plasticity and place cell adaptation, is largely unexplored.

Hippocampal interneurons in CA1 are diverse and sub-categorised by genetic, morphological and physiological properties[14–18]. Major sub-categories are parvalbumin (PV) expressing interneurons that are mostly perisomatic targeting and somatostatin (SOM) expressing interneurons that are mostly dendrite targeting[15,17], but within these general subtypes there is considerable heterogeneity[19]. Manipulation of all CA1 interneuron activity disrupts many features of place cell activity, including place cell formation[20–22]. Specifically, targeted silencing of PV interneurons alters the timing of place cell firing with respect to theta cycles whereas SOM interneuron silencing enhances place cell burst firing[23]. The dendritic targeting of inhibition by SOM interneurons imparts them with privileged control of synaptic inputs, whereas perisomatic targeting of inhibition by PV interneurons provides them with fine control of pyramidal cell spike timing and rate[24]. This suggests that SOM interneurons are prime candidates to regulate synaptic integration, plateau potentials, and thus the plasticity associated with place cell formation and remapping.

Activity of different interneuron subtypes is also associated with behavioural state induced by experience. In novel environments, where plasticity enables place cells to remap to the new environment, interneurons increase or decrease their activity in line with their subtype: PV interneurons generally increase activity and SOM interneurons decrease firing rate, but with considerable variability perhaps reflecting distinct subtypes within the PV or SOM classification (see refs. 25–29). Within the family of CA1 SOM interneurons, oriens lacunosum moleculare (OLM) interneurons have a unique morphology, receiving

School of Physiology, Pharmacology and Neuroscience, University Walk, University of Bristol, Bristol, UK. ✉e-mail: Jack.Mellor@bristol.ac.uk

feedback excitation from CA1 in stratum oriens (SO) and targeting inhibition specifically to the distal dendrites of CA1 pyramidal cells in stratum lacunosum moleculare (SLM). They also receive direct cholinergic input from the medial septum and express nicotinic and muscarinic receptors that respond to the release of acetylcholine[30–33]. Indeed, the majority of OLM interneurons express the α2 nicotinic receptor (OLMα2[33,34]), which can be used to selectively manipulate and measure OLMα2 interneurons using Chrna2-cre mice[31]. Surprisingly, stimulation of OLMα2 interneurons in ex vivo slices can enhance synaptic plasticity at CA3-CA1 Schaffer collateral (SC) synapses via a disinhibitory action on PV interneurons[31], but the role of OLMα2 interneurons in regulating plasticity driven by entorhinal inputs and plateau potentials has not been investigated. Furthermore, disruption of OLMα2 activity impairs hippocampal-dependent learning in novel object recognition, y-maze and fear conditioning paradigms[32,35–37], but the contribution of OLM interneurons to the plasticity of hippocampal neuronal representations required for learning is unknown.

Here, we made use of Chrna2-cre mice to selectively measure and manipulate OLMα2 interneurons in the hippocampus. By optogenetically stimulating OLM interneurons in ex vivo slices, we show that they control dendritic $Ca^{2+}$ signalling and synaptic plasticity initiated by entorhinal inputs to CA1 pyramidal neurons. Measurement of OLM activity during behaviour revealed a strong regulation by novelty and movement, where OLM firing was reduced in a novel environment that requires place cell remapping. Finally, bidirectional manipulation of OLM activity at specific locations within novel and familiar environments showed that OLM interneurons regulate the emergence and stability of new place cell representations. Together, these results demonstrate a critical role for dendrite-targeted inhibition for control of synaptic plasticity that underpins the remapping of neuronal representations necessary for adapting to unfamiliar environments.

## Results
### Control of synaptic plasticity and dendritic calcium dynamics by OLM interneurons

Behaviourally relevant synaptic plasticity that underlies place cell adaptations in CA1 is thought to be induced by coincident CA3 and entorhinal cortex inputs that summate supra-linearly and generate large dendritic $Ca^{2+}$ events termed plateau potentials[6,7,9,38,39]. The precise targeting of OLM interneuron synapses to the distal dendrites of CA1 pyramidal cells, where entorhinal cortex inputs are located, suggests they may have an important role in the integration of entorhinal and CA3 inputs and therefore the generation of behaviourally relevant synaptic plasticity. To test this, we selectively activated a subset of OLM interneurons that express Chrna2 by expressing the light-activated cation channel channelrhodopsin-2 (ChR2) in a cre-dependent manner using mice that expressed cre recombinase under control of the promoter for Chrna2 (Chrna2-cre) crossed with mice expressing cre-dependent ChR2 (Ai32 mice; methods). Immunohistochemistry confirmed that Chrna2-cre mice expressed cre selectively in OLM interneurons with reporter expression highest in the SO and SLM layers and cell bodies principally located in SO (Fig. 1A)[31,40]. This expression profile is consistent with OLM interneurons providing distal dendritic inhibition that exhibits slower synaptic current kinetics when measured at the soma than perisomatic targeting PV expressing interneurons[41].

Using this approach to selectively control OLM interneurons, we investigated whether OLM interneurons can control behaviourally relevant synaptic plasticity. Excitatory postsynaptic currents (EPSCs) elicited in 3 independent pathways (SC control, SC test and temporoammonic (TA) test) were recorded from CA1 pyramidal neurons, via whole-cell patch clamp. Plateau potentials can be replicated in ex vivo slices by bursts of high-frequency synaptic input; therefore, to induce synaptic plasticity, coincident theta burst stimulation (TBS) was given to the SC and TA test pathways. Since OLM interneurons are feedback

interneurons excited by CA1 pyramidal neurons and active during the same phase of theta cycles in vivo, the impact of OLM input was assessed by concurrent TBS of OLM interneurons (Fig. 1B). In the absence of OLM stimulation LTP was induced in the SC test pathway but not in the TA test pathway (Fig. 1C; 2.22 ± 0.59 (SC), 1.39 ± 0.44 (TA), $n = 7$). In contrast, in the presence of OLM stimulation SC LTP was blocked and instead LTD emerged in the TA test pathway (Fig. 1D; 0.94 ± 0.22 (SC), 0.46 ± 0.12 (TA), $n = 7$). This indicates that when synaptic plasticity is induced by a behaviourally relevant synaptic stimulation, OLM interneurons inhibit LTP at SC synapses, in contrast to classically induced high frequency stimulation to only SC fibres.

Plateau potentials in CA1 pyramidal neurons generate large $Ca^{2+}$ increases in SR apical dendrites, which triggers synaptic plasticity at SC synapses, but OLM interneurons target SLM distal dendrites. Therefore, we next tested whether OLM interneurons can regulate $Ca^{2+}$ in SR dendrites during plateau potentials. To measure dendritic $Ca^{2+}$ fluctuations, $Ca^{2+}$ dynamics were imaged in ex vivo slices using 2-photon imaging of defined apical SR and SLM dendritic regions during electrical synaptic stimulation of SC and/or TA inputs. CA1 pyramidal neurons were filled with the morphological marker Alexa594 and the $Ca^{2+}$ indicator Fluo5f and dendritic $Ca^{2+}$ and somatic voltage responses to synaptic stimulation recorded (Fig. 2A). Coincident TBS of TA and SC inputs that induced LTP in SC synapses generated large depolarisations at the soma that lasted for an extended period (>200 ms) similar to plateau potentials that induce new place fields in vivo[6–13]. These depolarisations were associated with large and long-lasting dendritic $Ca^{2+}$ increases in the SR region of apical dendrites (Fig. 2A). Concurrent TBS of OLM interneurons substantially reduced the SR dendritic $Ca^{2+}$ signal without any impact on the somatically recorded depolarisation (Fig. 2B; 62.28 ± 6.81% ($Ca^{2+}$), 99.41 ± 8.01% (depolarisation)). The lack of effect of OLM interneuron stimulation on somatic depolarisation was true for both multiple trains of stimuli during LTP induction (Fig. S1) and individual bursts of theta frequency stimulation (Fig. 2B) and is reminiscent of other situations where dendritic and synaptic voltage dynamics are localised and do not propagate linearly to the soma[39,42,43]. In comparison, when SC inputs were stimulated in the absence of TA stimulation, smaller depolarisations and $Ca^{2+}$ increases were generated and no effect of coincident OLM interneuron stimulation was observed (Fig. 2B; 116.3 ± 25.59% ($Ca^{2+}$), 108.6 ± 26.69% (depolarisation)). This indicates that OLM interneurons can reduce the $Ca^{2+}$ response in SR dendrites by preventing the integration of coincident TA inputs.

To further test this conclusion, we spatially dissected the action of OLM interneurons on dendritic $Ca^{2+}$ by stimulating SC and TA inputs separately and measuring the impact of OLM interneuron stimulation on dendritic $Ca^{2+}$ in apical SR and SLM regions. In these experiments, dendritic $Ca^{2+}$ increases were induced by increasing the stimulation intensity for each input pathway in a stepwise manner. OLM stimulation decreased $Ca^{2+}$ transients in the distal SLM dendrites regardless of the site of synaptic stimulation whereas $Ca^{2+}$ transients were only decreased at proximal SR dendrites when TA inputs were stimulated and not when SC inputs were stimulated alone (Fig. 2C–F; SLM/SC – $F_{(1, 6)} = 20.46$, $P = 0.004$, SLM/TA – $F_{(1, 5)} = 8.3$, $P = 0.035$, SR/TA – $F_{(1, 5)}$, $P = 0.02$, SR/SC – $F_{(1, 5)} = 0.39$, $P = 0.56$). This demonstrates that OLM interneurons control not only the $Ca^{2+}$ transients at distal dendrites but also $Ca^{2+}$ transients in proximal dendrites if the depolarisation originates in distal dendrites. This provides an explanation for how OLM interneurons control SR dendrite $Ca^{2+}$ during plateau potentials and, therefore, synaptic plasticity at SC synapses on apical CA1 dendrites.

### OLM activity during exploration in novel environments
Synaptic plasticity leading to place cell formation and remapping is predicted to occur more frequently when mice explore novel environments[10,44]. The question we next addressed is whether OLM

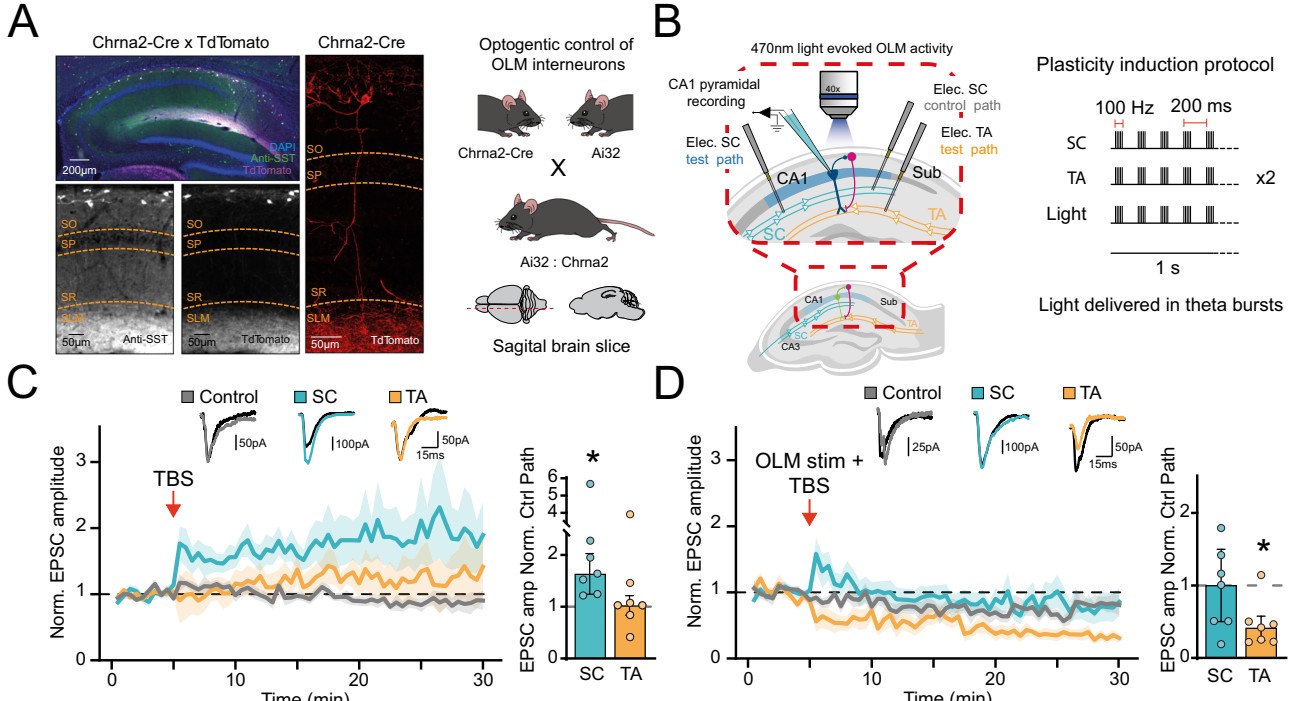

**Fig. 1 | OLM neurons inhibit associative synaptic plasticity in CA1 pyramidal neurons. A** Immunohistochemistry showing expression of somatostatin (SST) and tdTomato in Chrna2-cre x loxP-stop-loxP-tdTomato mice in different hippocampal layers: Stratum Oriens (SO), Stratum Pyramidal (SP) Stratum Radiatum (SR) and Stratum Lacunosum Moleculare (SLM). **B** Schematic of experimental approach. Ex vivo hippocampal slices prepared from Chrna2-cre x loxP-stop-loxP-ChR2 mice and recordings made from CA1 pyramidal cells whilst stimulating Schaffer collateral (SC), temporoammonic (TA) and OLM inputs. Plasticity protocol shown on right with or without light stimulation of OLM interneurons. **C** Theta Burst Stimulation (TBS) delivered concurrently to SC and TA inputs induced LTP at test pathway SC but not TA synapses. Example traces before and after plasticity shown above ($P = 0.0156$, paired $t$-test, two-tailed, $n = 7$ cells from 6 animals). Data presented as mean values ± S.E.M. **D** Same as C but with light stimulation of OLM interneurons during TBS. No LTP induced in SC test pathway and LTD induced in TA pathway ($P = 0.0312$, paired $t$-test, two-tailed, $n = 7$ cells from 7 animals). Data presented as mean values ± S.E.M.

interneurons play a role in enabling synaptic plasticity in novel environments by altering their firing rate. To answer this question, OLM activity was monitored in vivo by measuring $Ca^{2+}$ dynamics in OLM interneurons using miniaturised microscopy during exploration of familiar and novel environments. AAV1-Flex-GCaMP6s was injected into the dorsal hippocampus of Chrna2-cre mice and a GRIN lens and baseplate implanted above the hippocampus to selectively express and image the $Ca^{2+}$ indicator GCaMP6s in OLMα2 interneurons (Fig. 3A). Mice were trained to run from end to end of a 140 cm linear track for 10% sucrose rewards whilst tethered to a miniscope camera that recorded the activity of OLM interneurons (Fig. 3B). To investigate the response of these interneurons to novel environments OLM activity was first observed during exploration of a familiar environment for 10 min before switching the mice to a novel environment. The average OLM activity on the initial laps exploring a novel environment was significantly lower than the activity of the latter laps of exploration in a familiar environment ($0.011 ± 0.005$ vs $0.093 ± 0.063$ $\Delta F/F_0$). In comparison, mice transitioned between two exposures of the familiar environment showed no change in OLM activity ($0.084 ± 0.008$ vs $0.082 ± 0.008$ $F/F_0$) (Fig. 3C,D). Analysis of single interneurons revealed the majority of interneurons reduced their activity in the novel environment and this reduction was more prominent for interneurons that were more active in the familiar environment (Fig. 3E). As expected during novel exploration the velocity of the mice was reduced, as the mice explored unfamiliar surroundings (Fig. 3F and Fig. S2A) and similar to other interneuron populations within the hippocampus[27,29,45] we observed that OLM activity was also correlated with velocity of the mice (Fig. 3B, G, H). Therefore, it was important to distinguish between any reduction in OLM activity due to a change in velocity and changes in OLM activity due to a novelty signal. To do this

the average activity of the interneurons was binned based on the velocity of the mice. This revealed that at comparable velocities, the activity of OLM interneurons was consistently reduced (Fig. 3G). The correlation between interneuron activity and mouse velocity was still observed in the novel environment, although significantly reduced compared to familiar exploration (Fig. 3G, H). These observations suggest that OLM activity is positively modulated by mouse velocity and negatively modulated by novelty, suggesting that synaptic plasticity at CA1 pyramidal neurons is more likely to occur in novel environments when place cells are remapped.

## Place cell formation and remapping are modulated by OLM interneuron activity

With OLM interneurons reducing their activity in novel environments and able to modulate excitatory synaptic plasticity, we hypothesised that OLM interneurons may play a key role in the mechanisms of place cell formation and the plasticity that leads to their long-term stability[46]. To test this, we recorded place cell formation and stability in vivo whilst manipulating the activity of OLM interneurons using miniaturised microscopy. To express the $Ca^{2+}$ indicator GCaMP6f into dorsal hippocampal CA1 pyramidal neurons, we injected Chrna2-cre mice with AAV5-CAMKII-GCaMP6f alongside either AAV5-hSyn-Flex-ChrimsonR, AAV5-hSyn-Flex-Jaws or the control AAV5-hsyn-DIO-mCherry to express either the redshifted excitatory opsin ChrimsonR or the redshifted inhibitory opsin Jaws in OLMα2 interneurons. This enabled us to record the activity of pyramidal neurons whilst either stimulating or inhibiting the activity of OLM interneurons. OLM activity was modulated using 620 nm light through the miniscope when mice entered a specific 33 cm section of the track termed the 'optogenetic-zone' with an adjacent 33 cm section of the track without light delivery used as a

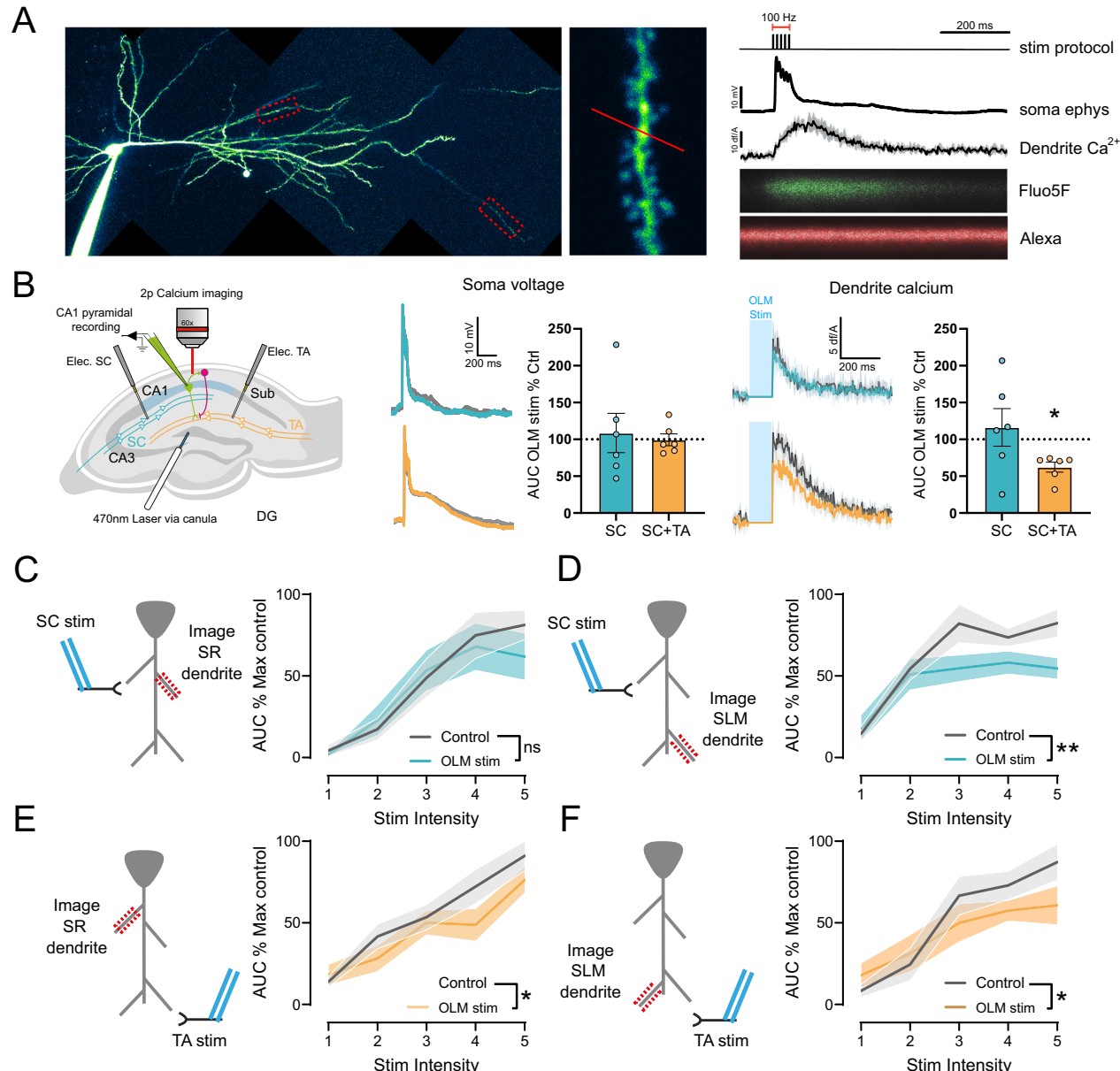

**Fig. 2 | OLM neurons inhibit dendritic Ca²⁺ during plateau potentials. A** CA1 pyramidal cell filled with Alexa594 and Fluo5. Recorded SR and SLM regions of the dendrite are highlighted with red dashed boxes, with a high magnification image of the line scan region (middle). Left, example recording from illustrated line scan with dF/A quantified trace and somatically recorded voltage in response to concurrent stimulation of SC and TA inputs. **B** OLM stimulation reduces SR dendritic Ca²⁺ in response to concurrent TA and SC inputs but not when only SC inputs are stimulated. Example somatic EPSPs and SR dendrite Ca²⁺ responses are shown with and without OLM stimulation. Note blanked region of Ca²⁺ response during light stimulation of OLM interneurons ($P = 0.0312$, Wilcoxon matched-pairs signed rank test, two-tailed, $n = 6$ cells from 4 animals). Increasing stimulation intensity to SC (**C, D**) or TA (**E, F**) input pathways increases Ca²⁺ responses in SR (**C, E**) or SLM (**D, F**) dendritic regions. OLM stimulation reduces dendritic Ca²⁺ responses in (**D, E, F**) but not (**C**). SR/SC (**C**) − F(1, 5) = 0.39, $P = 0.56$, SLM/SC (**D**) − F(1, 6) = 20.46, $P = 0.004$, SR/TA (**E**) − F(1, 5), $P = 0.02$, SLM/TA (**F**) - F(1, 5) = 8.3, $P = 0.035$, Mixed-effects model (REML), $n = 6$ (4), 7 (4), 6 (4), 6 (5) For **C, D, E, F** respectively ($n$ listed as: cell (animal)). Data presented as mean values ± S.E.M.

direct comparator termed the 'control zone' (Fig. 4A). Importantly, within experiment comparisons between optogenetic and control zones provided a control for potential excitation of ChrimsonR or Jaws by 455 nm light used for Ca²⁺ imaging.

Analysis of CA1 pyramidal neuron Ca²⁺ activity whilst mice traversed a familiar linear track revealed place cell firing fields that tiled the entire length of the track. When mice were switched to a novel track the spatial encoding of these cells was completely lost and a new spatial representation was formed from a population of de novo place cells (Fig. 4B). The switch to a novel track caused an increase in overall pyramidal cell activity (20 ± 5%, $n = 13$, $p < 0.01$) which subsided as the

track became familiar[10,47] (Fig. 4G and Fig. S2C). To assess the impact of OLM activity on place cell formation and remapping, pyramidal neuron activity was recorded during exploration in a novel environment (session 1) with and without optogenetic manipulation of OLM interneurons. To subsequently assess the stability of these newly formed place cells and the impact of OLM perturbation, mice were re-exposed to the same novel environment for a second time (session 2) with a gap of 3 min between sessions and place cell representations were compared between Session 1 and Session 2.

Increased OLM activity by activation of chrimsonR in the optogenetic stimulation zone of the novel track did not alter mouse

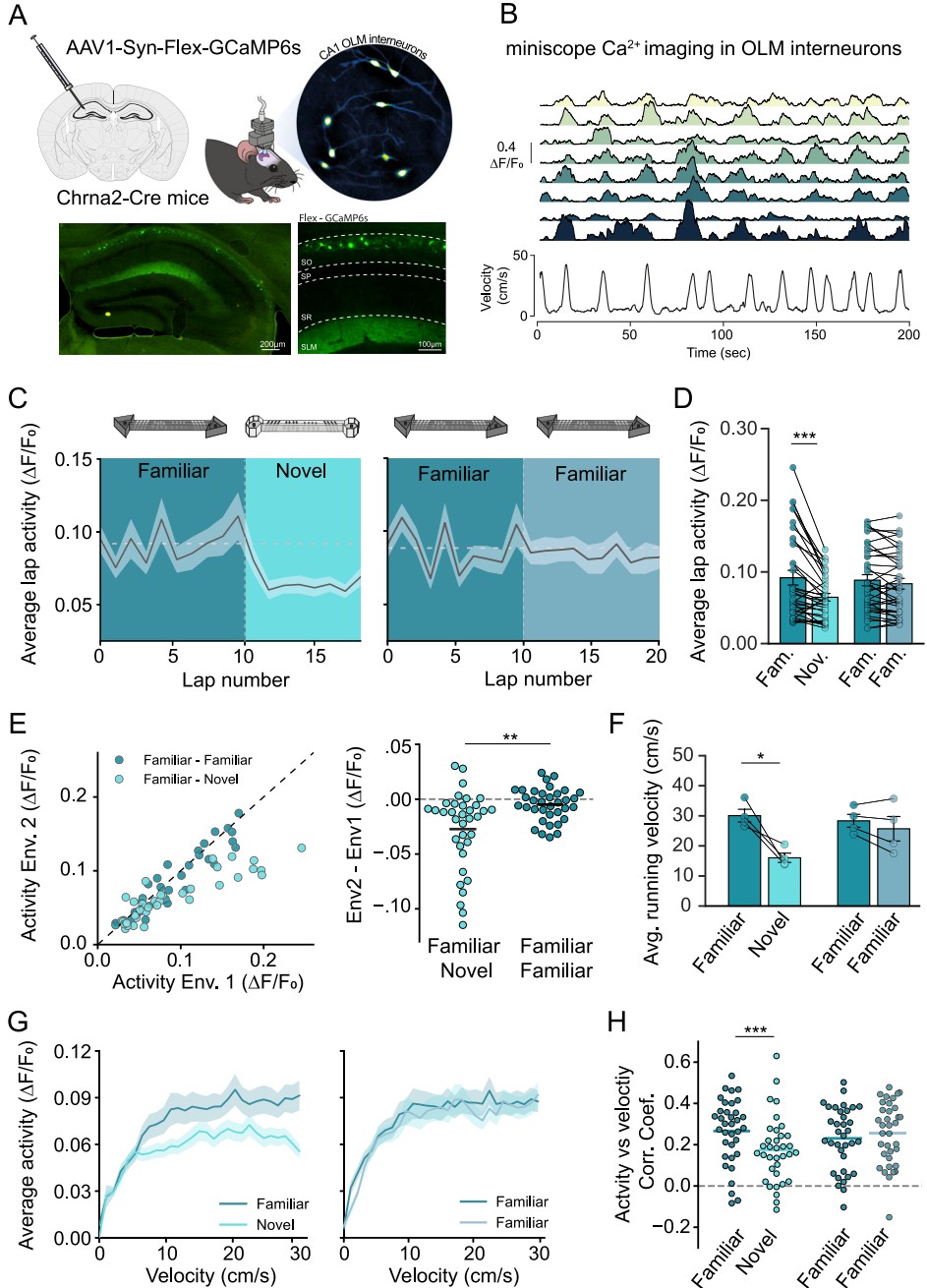

**Fig. 3 | OLM activity reduces in novel environments. A** Chrna2-cre mice were injected in the dorsal hippocampus with a CRE dependent viral construct to express GCaMP6s in OLM interneurons. Miniscope field of view of GCaMP6s expression in OLM interneurons. Widefield fluorescence of the dorsal hippocampus brain slice showing expression of GCaMP6s restricted to OLM interneurons. **B** Example Ca²⁺ fluorescence recording from 8 OLM interneurons within the same mouse together with the corresponding mouse velocity as the mouse explored a familiar environment. **C** Average lap-by-lap activity of OLM interneurons as mice explored a familiar and then a novel environment (left) and two consecutive familiar environments (right). **D** Average activity of individual OLM interneurons as they explored different environments. Average activity across all laps $n = 35$ cells (from 4 mice).

Paired $t$-test (two-tailed) ***: $p = 0.00013$. **E** Average activity of each interneuron in familiar environment vs average activity in the subsequent environment (novel or familiar) (left). Activity difference between the two environments, $n = 35$ cells (from 4 mice). Unpaired $t$-test (two-tailed) **: $p = 0.0012$. **F** Average running velocity of mice during exploration of familiar and novel environments. $n = 4$ mice, paired $t$-test (two-tailed) *: $p = 0.023$. **G** Average OLM interneuron activity at different mouse velocities during exploration of familiar and novel environments. **H** Individual OLM interneuron activity vs mouse velocity Pearsons correlation coefficients between familiar and novel environments and two consecutive exposures to a familiar environment. $n = 35$ cells (from 4 mice). Paired $t$-test (two-tailed) ***: $p = 0.0010$. Data presented as mean values ± S.E.M.

behaviour as assessed by velocity in the opto zone (Fig. S2B) but reduced the proportion of de novo place cells that represented that spatial location (Fig. 4C, D). This corresponded with a reduction in the event rate of the pyramidal neurons specific to the region of the track where OLM activity was increased (Fig. 4G). Together with the lack of

any effects of LED stimulation in the mCherry expressing mice (Fig. S3), this confirms the predicted impact of enhanced inhibition on pyramidal neuron firing and demonstrates the efficacy of OLM interneuron optogenetic excitation (Fig. 4G). In contrast, reducing the activity of OLM interneurons with the inhibitory opsin Jaws caused a

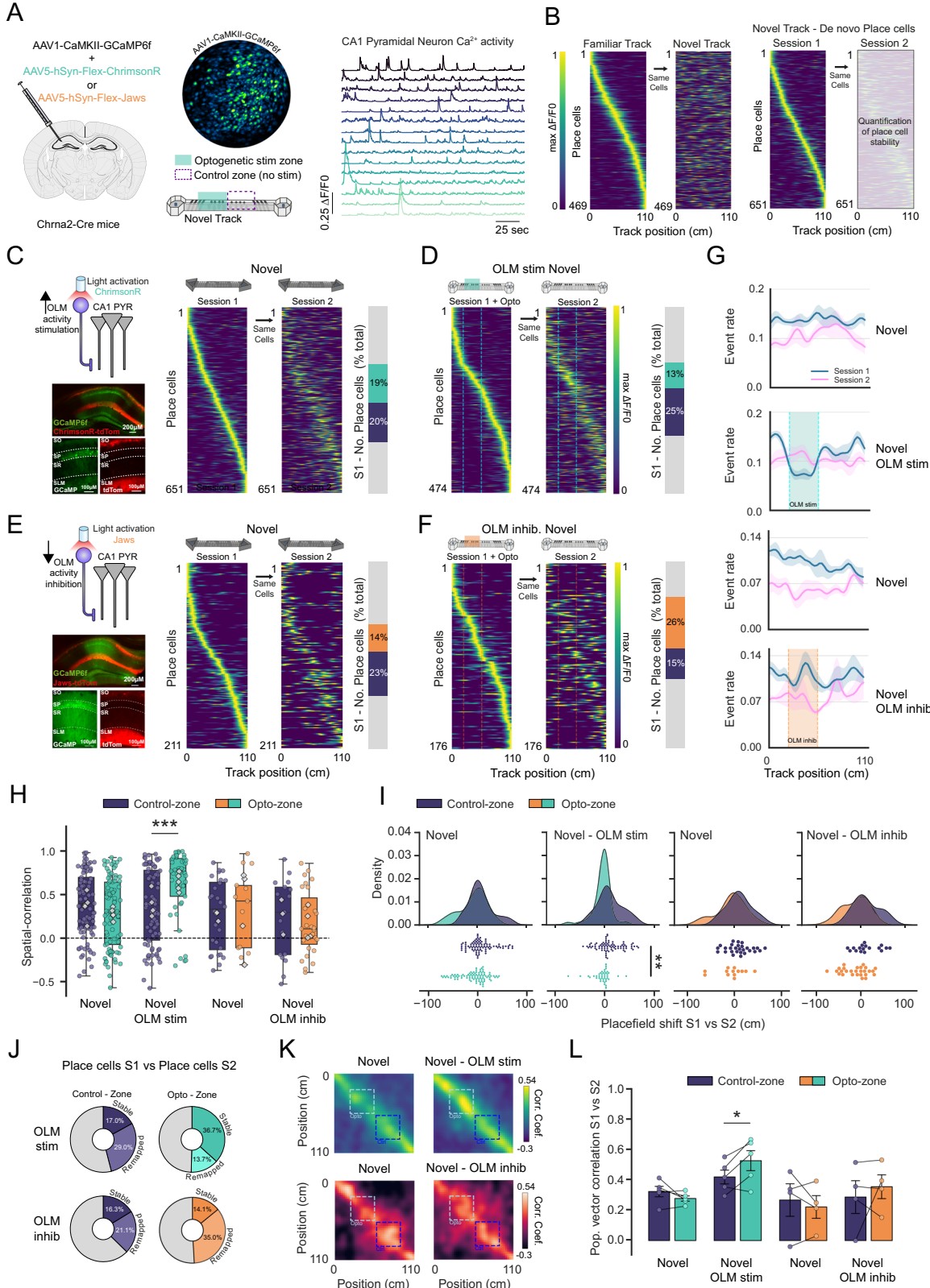

small and non-significant increase of pyramidal neuron activity that led to an overrepresentation of de novo place cells selective to the optogenetic stimulation zone (Fig. 4E–G).

To assess the stability of newly formed spatial representation, we analysed the stability of individual place cells between session 1 and session 2 in the novel environment. By comparing the place cells formed in session 1 with the place cells formed in session 2 we could

determine if modulating the activity of OLM interneurons lead to an increased retention of place cells encoding the same location in specific regions of the track. Increasing OLM activity during place cell formation led to a population of place cells with significantly higher spatial firing stability between the two sessions and place fields that shifted their location significantly less compared to place cells formed in the absence of increased OLM activity (Fig. 4H, I; 0.363 ± 0.049 vs

**Fig. 4 | OLM activity regulates stability of spatial representations in novel environments. A** Chrna2-cre mice were injected into the dorsal hippocampus with viral constructs to express GCaMP6f in pyramidal and either ChrimsonR or Jaws into OLM interneurons (left). Miniscope field of view of GCaMP6f expression in pyramidal neurons and example Ca²⁺ transients from recorded pyramidal neurons. Opsins were activated at a specific portion of the track (optogenetic stim zone), data were then compared to a control zone. **B** Place cell firing rate maps in a familiar environment with the same cells firing rates in a novel environment, demonstrating global remapping in a novel environment. De novo place cells were acquired during the exploration of the novel environment that tiled the full length of the track. **C** Example expression of excitatory opsin ChrimsonR in OLM interneurons to increase the activity of OLM interneurons (left). De novo place cells acquired during the first exposure to the novel environment (session 1) in the absence of optogenetic stimulation. Activity of the same cells in session 1 during the second exposure to the novel environment (session 2) (middle). Proportions of place cells during session 1 encoding different zones on the track (right, grey depicts zones at the ends of the track). **D** De novo place cells acquired in the first exposure to a novel environment (session 1) with optogenetic activation of ChrimsonR to stimulate OLM interneuron activity. Activity of the same cells in session 1 during the second exposure to the novel environment (session 2) (left). Proportions of place cells during session 1 encoding different zones on the track (right, grey depicts zones at the ends of the track). **E** Same as **C** except for the expression of Jaws opsin to

decrease activity in OLM interneurons. **F** Same as **D** except for optogenetic activation of Jaws to decrease OLM interneuron activity. **G** Ca²⁺ event rate of all recorded neurons along the length of the track during the exploration of novel environments (OLM stim (top row) $n = 5$ mice, opto vs control zones $p = 0.018$ (paired $t$-test, two-tailed), OLM inhib. (bottom row) $n = 4$ mice, opto vs control zones $p = 0.2$ (paired $t$-test, two-tailed)). **H** Spatial correlation values between place cells in session 1 and the activity of the same cells in session 2, for both control and optogenetic zones. Filled symbols are cells, open symbols are averages for each animal. Unpaired $t$-test with $n =$ cells on Fisher corrected data ***: $p = 0.0007$. Data presented as box plots (median, interquartile range) plus whiskers (1.5×interquartile range). **I** Place field location shifts from place cells in session 1 vs place field location of the same cells in session 2. Unpaired $t$-test on absolute position shifts **: $p = 0.001$. **J** Retention of session 1 place cells during session 2, place cells were either lost, and no longer classified as place cells (grey), stable places encoding a similar location in session 2 or remapped place cells encoding a new location in session 2. Average % across animals. **K** Population vector correlation matrix between activity rate maps in session 1 vs session 2 for (**C**–**F**), averaged across animals. **L** Average population vector correlation taken as average diagonal correlation for each track zone, (Control-zone vs Opto-zone). (OLM stim $n = 5$ mice, OLM inhib. $n = 4$ mice, (*: $p = 0.049$ paired $t$-test). Data presented as mean values ± S.E.M.

---

$0.634 \pm 0.049$ spatial correlation, $11.016 \pm 2.854$ cm vs $-4.231 \pm 2.621$ cm place field shifts). Inhibiting OLM activity during place cell formation did not significantly alter the place cells' spatial stability or alter the degree of place field location changes, but there was a trend towards lower spatial firing stability and location shifting (Fig. 4H, I). Importantly, neither stimulation nor inhibition of OLM interneurons resulted in de novo place cells that were overall more or less likely to be retained in the next session (Fig. 4J; $p > 0.05$) indicating that the impact of OLM manipulation is selective for place cells active during the manipulation and OLM interneuron manipulation did not alter place cell spatial information content or place field widths (Fig. S5A, B). Therefore, OLM activity not only regulates the number of de novo place cells that are formed but also the population of place cells with high spatial stability. This suggests that increased OLM activity reduces the proportion of unstable place cells, whilst reduced OLM activity leads to an increased proportion of unstable place cells. By categorising the place cells into stable and unstable based on a spatial correlation threshold of 0.5, we observe this to be the case with OLM activity bidirectionally altering the proportion of unstable place cells (Fig. S6C, D).

Next, we assessed the impact of the change in individual place cell stability on the population coding of position. The population vector correlations between the two exposures to the same novel environment (session 1 vs session 2) revealed that increasing OLM activity led to enhanced stability of spatial encoding restricted to the optogenetic stimulation zone ($0.421 \pm 0.046$ vs $0.529 \pm 0.066$ (correlations for control vs opto zone)). Interestingly, reducing the activity of OLM interneurons did not lead to a decrease in the population encoding of the track (Fig. 4K, L). This analysis indicates that the stabilisation of individual place cells in a novel environment by increasing OLM activity translates into a stabilisation of the population coding of position.

Place cell representations completely remap in a novel environment, but in the hippocampus, there is also continual refinement of spatial representations even in familiar environments[48]. To investigate whether OLM interneurons also regulate place cell stability in familiar environments or whether their role is selective for novel environments, we next manipulated OLM activity during exploration of a familiar environment. In agreement with previous studies[48], the stability of place cell representations between sessions on a familiar track was notably higher than those of newly learnt representations but still resulted in measurable drift (Fig. 5A, C, G, H). Similar to the

experiments in novel environments, OLM activity was selectively increased or decreased via optogenetic activation of ChrimsonR or Jaws on a defined section of the track. The acute effects of manipulating OLM activity were similar to that found in novel environments; increasing OLM activity reduced the number of place cells representing the stimulated region of the track, whereas decreasing OLM activity had a limited effect with a trend towards an increased number of place cells (10% vs 21% and 18% vs 16%). This again demonstrates that although OLM interneurons have the capacity to inhibit place cells, at basal firing rates in familiar or novel conditions, OLM interneurons do not appear to be greatly inhibiting place cell firing rates.

In a novel environment, models predict that reducing OLM activity facilitates place cell remapping and our data show that increasing OLM activity promotes stability (Fig. 4). In a familiar environment neither decreasing nor increasing OLM activity altered place cell stability, with no changes in spatial correlations or the population vector, alongside no changes in place cell information content or place field widths (Fig. 5F–H, Fig. S5C, D). These findings suggest that OLM interneuron regulation of place cell stability is contingent on the conditions found during novel explorations.

Although OLM activity did not alter place cell stability, enhancing their activity was able to reduce the number of place cells encoding the stimulated region of the track. In a familiar environment, the reduction in place cell numbers may be due to silencing pre-existing place cells or reducing the rate of de novo place cell formation or remapping. To investigate the degree to which increased OLM activity may be silencing pre-existing place cells, we measured the retention of pre-existing place cells from a previous exploration of the same familiar environment (session 0). Within the control zone, the number of place cells retained from session 0 gradually decreased in session 1 and session 2, characteristic of place cell representational drift.

Interestingly, OLM interneuron stimulation during session 1 led to a trend towards lower place cell retention compared to the corresponding control zone. In addition, place cell retention in the opto-zone was not further reduced in session 2, unlike place cells in the control zone that displayed a linear decrease in the place cell retention across sessions. (Fig. S5E, F). This suggests that in addition to the characteristic place cell representational drift, increased OLM activity during session 1 also silenced a proportion of session 0 place cells that subsequently returned in session 2. Conversely, OLM interneuron inhibition during session 1 slightly increased place cell retention compared to the corresponding control zone (Fig. S5H, I).

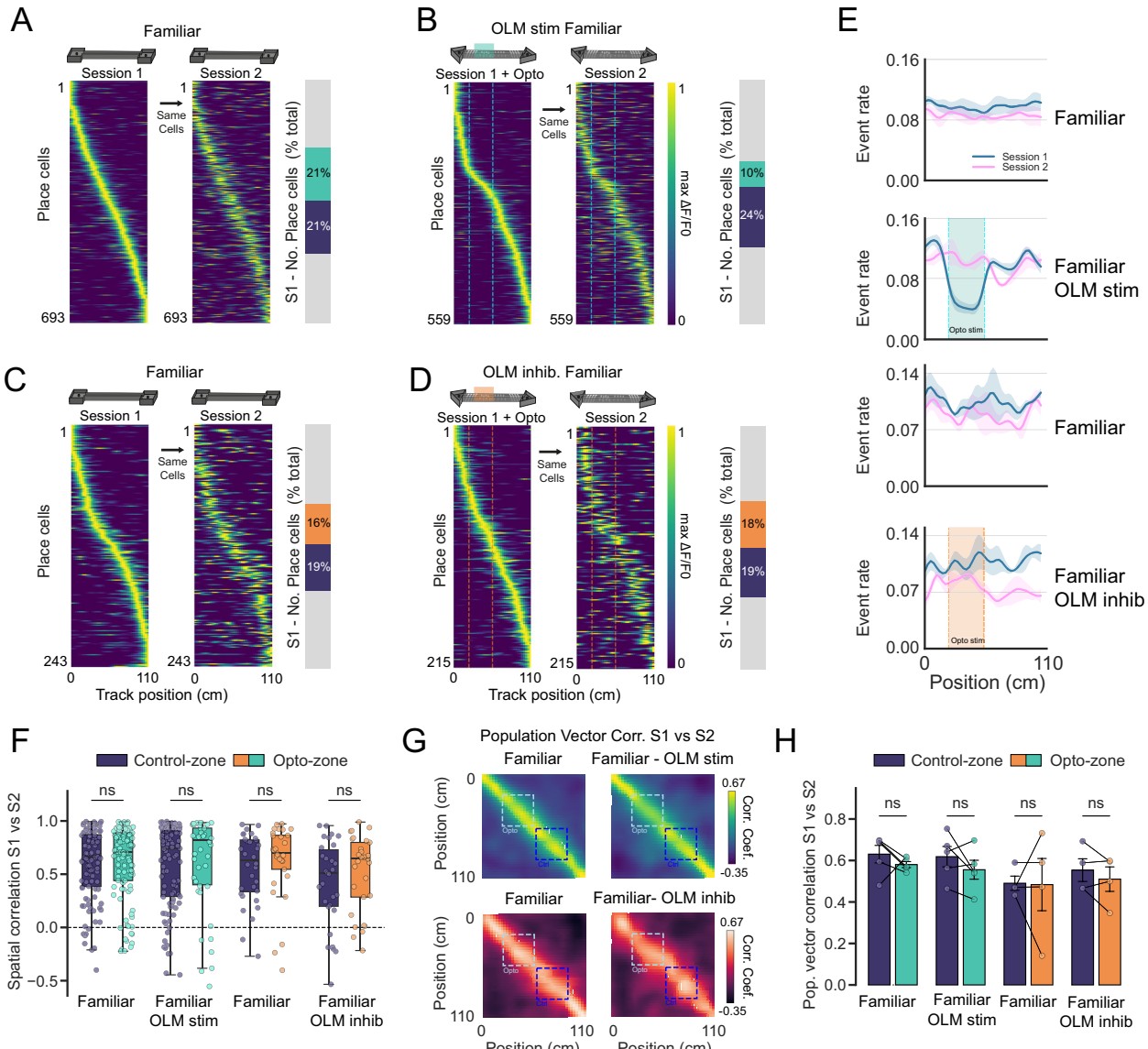

**Fig. 5 | Stability of spatial representations in familiar environments is unaltered by OLM activity. A** Place cell firing rate maps in a familiar environment during session 1 in the absence of optogenetic stimulation of excitatory opsin ChrimsonR. Activity of the same cells in session 1 during a second exposure to the familiar environment (session 2) (left). Proportions of place cells during session 1 encoding different zones on the track. **B** Place cell firing rate maps in a familiar environment with optogenetic stimulation of excitatory opsin ChrimsonR at a specific portion of the track (session 1). Activity of the same cells in session 1 during a second exposure to the familiar environment (session 2) (left). Proportions of place cells during session 1 encoding different zones on the track. **C** Same as A except for mice expressing the Jaws opsin in OLM interneurons. **D** Same as B except for optogenetic activation of Jaws to decrease OLM interneuron activity during familiar exploration. **E** Ca²⁺ event rate of all recorded neurons along the length of the track during the

exploration of familiar environments (OLM stim (top row) $n = 5$ mice opto vs control zones $p = 0.013$ (paired $t$-test, two-tailed), OLM inhib. (bottom row) $n = 4$ mice, opto vs control zones $p = 0.6$ (paired $t$-test, two-tailed). **F** Spatial correlation values between place cells in session 1 and the activity of the same cells in session 2, for both control and optogenetic zones. Data presented as box plots (median, interquartile range) plus whiskers (1.5× interquartile range). Unpaired $t$-test (two-tailed) with $n$ = cells on Fisher corrected data. **G** Population vector correlation matrix between activity rate maps in session 1 vs session 2 for **C**–**F**, averaged across animals. **H** Average population vector correlation taken as average diagonal correlation for each track zone, (Control-zone vs Opto-zone). (OLM stim $n = 5$ mice, OLM inhib. $n = 4$ mice. Paired $t$-test (two-tailed). Data presented as mean values ± S.E.M.

To further investigate whether manipulating OLM activity altered the degree of de novo place cell formation in a familiar environment, we compared the place cells in session 1 to the same cells in session 0. We found that the percentage of de novo place cells (place cells in session 1 but not place cells in session 0), was not significantly changed in the optogenetic stimulation zone compared to the control zone for either OLM excitation or inhibition (Fig. S5G, J; OLM stimulation 18 ± 5% vs 23 ± 7% and OLM inhibition 49 ± 18% vs 21 ± 10%, opto vs control zone, $p > 0.05$). Taken together, these results indicate that the role of OLM interneurons in modulating place cell formation is

restricted to conditions in which the mouse experiences a novel environment.

## OLM interneuron activity prolongs the lifetime of novel environment representations

Hippocampal place cell representations of familiar environments drift within and across days[48–51] but how this compares to the degree of drift that occurs for novel environments across days is not documented[52]. Furthermore, our results indicate that representations of novel environments drift faster than representations of familiar environments, so

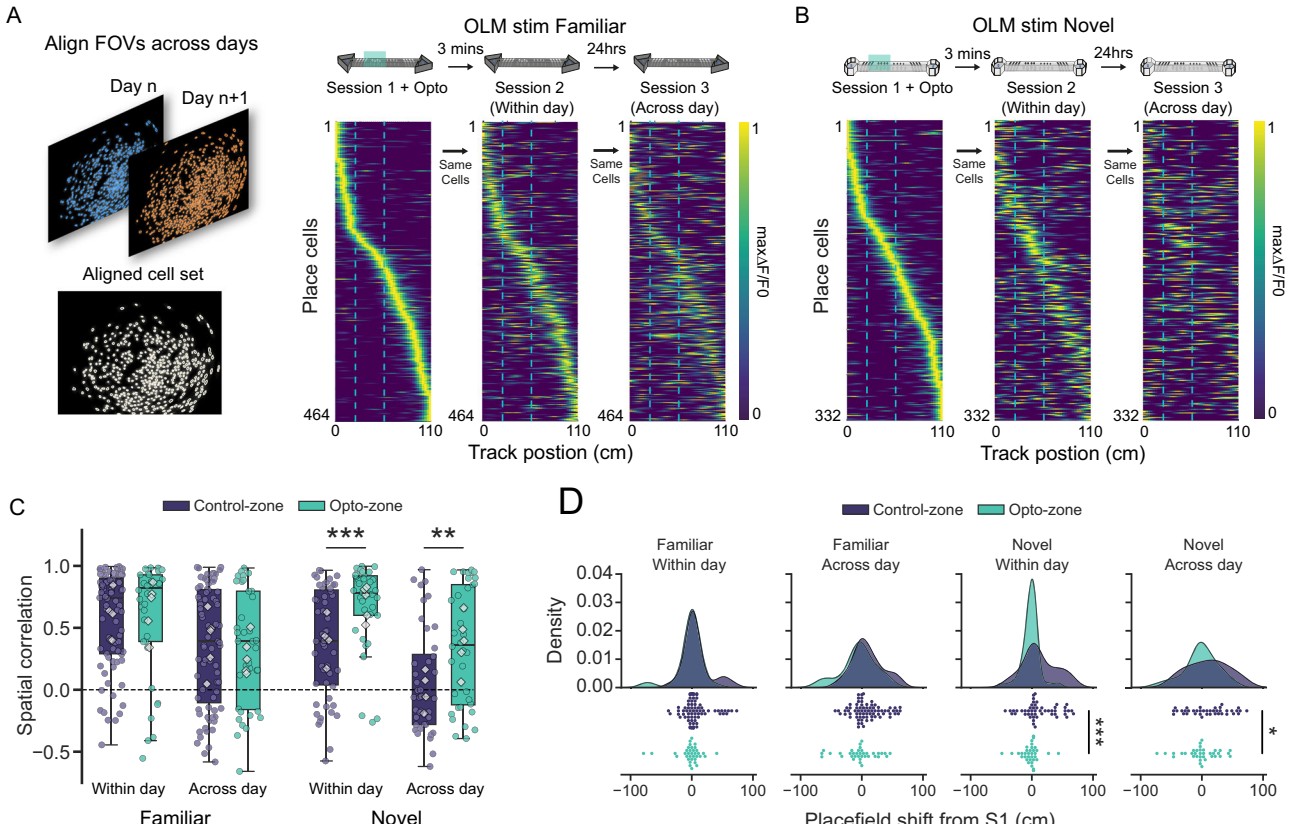

**Fig. 6 | Stabilised spatial representations are maintained for 24 h. A** Cells are aligned and registered across recording sessions. Place cell firing rate maps in a familiar environment with optogenetic stimulation of excitatory opsin ChrimsonR at specific portion of the track (session 1). Activity of the same cells in session 1 during a second exposure to the familiar environment after a 3 min gap (session 2, within day, middle) or 24 h later (session 3, across day, right). **B** Place cell firing rate maps as in (**A**), but for a novel environment. **C** Spatial correlation values between place cells in session 1 and the activity of the same cells in session 2 or session 3, for both control and optogenetic zones. Filled symbols are cells, open symbols are averages for each of 5 animals. $n$ = 77, 36, 75, 36, 51, 33, 40, 36 cells (left to right, from 5 mice) Unpaired $t$-test (two-tailed) with $n$ = cells on Fisher corrected data **: $p$ = 0.0052, ***: $p$ = 0.00028. Data presented as box plots (median, interquartile range) plus whiskers (1.5× interquartile range). **D** Place field location shifts from place cells in session 1 vs place field location of same cells in session 2. Unpaired $t$-test (two-tailed) on absolute position shifts *: $p$ = 0.035, ***: $p$ = 0.0003.

we next asked whether stabilisation of place cells in novel environments by OLM interneuron activity could arrest this rapid representational drift. To address this, we added an additional recording session 24 h after the first recording session for both novel and familiar environments and registered individual cells across sessions both within and across days to ensure we were recording from the same cells for all sessions.

In the familiar environment, spatial correlation for individual place cells in the control zone of the track drifted across days, but there remained a degree of correlation after 24 h (Fig. 6B, C). In contrast, in the novel environment, spatial correlation in the control zone drifted substantially over 24 h such that there was little correlation remaining (Fig. 6A, C). Remarkably, stimulation of OLM interneurons in the stimulation zone restricted the decline of spatial correlation in the novel environment to a similar extent found in the familiar environment, and this was true at both within day and across day timepoints (Fig. 6C). These results were mirrored by analysis of the place field shifts showing much more stability for place fields in the stimulated zone compared to the control zone for both within and across day sessions (Fig. 6D). The spatial correlation across days in the novel environment was on average 0 for the control zone, indicating a substantial loss of the newly learnt spatial representation. The rescued spatial correlation in the optogenetic stimulation zone suggests that OLM interneuron stimulation during novelty prevented the loss of the spatial representation across days, resulting in a more stable

representation with a degree of representational drift comparable to well-established familiar environments.

## Discussion

Synaptic plasticity and therefore remapping of place cells in CA1 is thought to be induced by large dendritic $Ca^{2+}$ transients driven by strong input from TA fibres from the entorhinal cortex coupled to input from CA3[6–13,38,39]. Here we show that OLM interneurons specifically inhibit this entorhinal input thereby reducing apical dendritic $Ca^{2+}$ and synaptic plasticity during associative plasticity induced by pairing both inputs (Figs. 1 and 2). This demonstrates the importance of OLM interneurons for gating behaviourally relevant synaptic plasticity. We then show that OLM activity is reduced at exactly the timepoint where synaptic plasticity is required to remap place cells—the experience of a novel environment (Fig. 3). In a final set of experiments, we test the impact of manipulating OLM activity on place cell formation and stability in novel and familiar environments given the prediction that OLM inputs gate behaviourally relevant synaptic plasticity. These experiments show that artificially reinstating OLM activity in a novel environment reduces the emergence of new place cells, but that the place cells that do emerge are more stable, in effect, pruning out the more 'speculative' place cells. In contrast, further reduction of OLM activity in a novel environment increased the proportion of unstable place cells (Figs. 4 and S6). In a familiar environment, increasing OLM activity reduced place cell firing rates but there was no substantially enhanced

stability of the remaining place cells (Fig. 5) demonstrating that reduced firing rates per se cannot explain the enhanced stability of place cells when OLM interneurons are stimulated in a novel environment. Furthermore, enhancing the stability of place cells in the novel environment meant that across days place cells retained some spatial correlation as opposed to losing almost all correlation (Fig. 6). For experiments examining place cell stability the use of control and OLM manipulation zones enabled us to make powerful within experiment comparisons. Overall, these results support the conclusion that OLM interneurons reduce synaptic plasticity and thereby stabilise place cells and spatial representations in the hippocampus. In a novel environment, OLM activity reduces to allow remapping of place cells.

We have previously demonstrated that inhibitory synapses from SOM interneurons (that include OLMα2 interneurons) to pyramidal neurons undergo potentiation when synaptic activation is paired with pyramidal neuron depolarisation[41]. By modelling the activity of place cells in vivo this inhibitory plasticity is predicted to regulate the stability of place cells in novel and familiar environments[41,46]. Interestingly, here we show that pairing of TA input with plateau potentials in CA1 pyramidal cells resulted in LTD of the TA input (Fig. 1D) and therefore that plateau potentials cause a simultaneous increase in inhibition and decrease in excitation for synaptic transmission to distal CA1 dendrites. Under this framework, the burst firing of place cells in their place fields drives potentiation of OLM synapses and depression of TA synapses, which in turn makes further plasticity of those place cells less likely in response to ongoing place cell activity—effectively stabilising the place cells. When OLM interneurons reduce their activity in novel environments less inhibitory plasticity is induced and therefore place cells can undergo more plasticity and exhibit less stability after remapping. This is indeed what we and other studies have found (c.f. Figs. 4C and 5A)[48,51,53]. Conversely, when we stimulate OLMα2 neurons in a novel environment, there will be more inhibitory plasticity, at least in the place cells that are still active, making those place cells more stable. This is what we see in Fig. 4 and therefore our results support these predictions for the role of inhibitory plasticity in place cell stabilisation.

An alternative explanation for the observed enhancement of place cell stability by OLM stimulation is that OLM activation specifically inhibits place cells that are more unstable leaving a population of stable place cells. Three pieces of data argue against this interpretation. Firstly, the proportion of stable place cells in the novel environment as a fraction of the total cells, and not just relative to unstable place cells, increases (Fig. 4J). Secondly, in the familiar environment when place cells are compared between sessions prior to and during OLM stimulation, stable, unstable and novel place cells are all reduced equally in the stimulation and control zones of the track (Fig. 5K). Thirdly, artificially reducing the spike rate in the mCherry dataset by randomly removing 30% of spikes in the opto stim zone to mimic the effect of OLM interneuron stimulation produced a similar reduction in the number of detected place cells but failed to increase stabilisation of the remaining place cells (Fig. S4). Therefore, the reduction in spike rate cannot explain the observed stabilisation of remaining place cells in the OLM stim track region.

The reduction in OLM activity when mice are placed in a novel environment is in line with reductions in SOM interneuron activity in novel environments[26–29] suggesting either that OLM interneurons (and specifically OLMα2 neurons) form the majority of observed SOM interneurons in CA1 or that a majority of SOM interneurons respond in a similar manner to novelty. There is some evidence for both conclusions. The SO location of the OLM cell bodies makes them more accessible to imaging than deeper SOM interneurons and therefore more likely to be recorded[26,45] but where SOM interneurons have been compared across different layers there does not appear to be a big difference in their response to novelty[29]. A significant proportion of OLM interneurons do not express nicotinic a2 receptors and target

their outputs more selectively to CA1 pyramidal neurons and not to fast spiking PV interneurons[34]. There is also evidence that the proportion of non-OLMα2 to OLMα2 interneurons is higher in dorsal CA1 compared to ventral[54]. We have not addressed the physiology of these non-OLMα2 OLM interneurons but the evidence suggests they respond to novelty and control dendritic excitability and synaptic plasticity in much the same way as OLMα2 interneurons[27–29]. The increase in OLM activity with velocity of movement also mirrors findings from SOM interneurons[26,27,29,45] although there is some heterogeneity[27,45]. This heterogeneity might be explained by the heterogeneity of SOM interneurons in general and OLM interneurons in particular[34] but we show that the reduced average velocity in novel environments does not account for the reduced OLM activity in response to novelty.

An important unanswered question is what controls the activity of OLM interneurons and specifically what causes reduced activity in novel environments? The main excitatory input to OLM interneurons is from CA1 pyramidal neurons, but their average activity inreases in a novel environment, so this is unlikely to cause the reduction of OLM activity. The major inhibitory inputs to OLM interneurons are from VIP expressing interneurons which strongly inhibit SOM interneurons in both hippocampal and neocortical circuits[15,29,55–59]. VIP expressing interneurons are themselves a diverse population of neurons that respond to inputs from hippocampal, cortical and subcortical areas and there are at least 2 subtypes that inhibit OLM interneurons. Interestingly, these subtypes are differentially modulated by animal velocity and therefore may be responsible for heterogeneity of OLM activity during movement[57]. Moreover, the activity of these VIP interneurons is increased in response to surprise or novelty, making them prime candidates to mediate the reduction of OLM activity with novelty[29,60,61]. Then the question is what drives VIP interneurons and their activity increase in novel environments? 2 primary sources have been identified: Intercortical inputs and neuromodulatory inputs from subcortical structures. In the hippocampus these are for example from entorhinal cortex and cholinergic projections from medial septum, respectively[15,62–64]. This suggests dual antagonistic roles for cholinergic input to the circuit since direct input to OLM interneurons excites OLM interneurons whereas indirect input to VIP interneurons will inhibit OLM interneurons[32,36,58,65]. In a behavioural context it appears that the cholinergic input to VIP interneurons is dominant since cholinergic inputs are disinhibitory[60,66,67] and during novelty cholinergic inputs are active leading to a reduction in OLMα2 activity (Fig. 3)[26,28,29]. However, during fear conditioning the aversive stimulus triggers cholinergic inputs and direct OLM activation[32].

SOM interneurons and OLM interneurons specifically are critical for the learning of new contexts and when hippocampal OLM or SOM interneurons are inhibited learning is impaired[32,35–37]. Overall, our data support a model where OLM activity is reduced in novel environments that require place cell representations to be remodelled by synaptic plasticity. The reduction in OLM activity is then a critical factor to provide a window for plasticity, and therefore place cell flexibility, that underpins learning.

## Methods

### Animal strains and breeding

All procedures and techniques were conducted in accordance with the UK animals scientific procedures act, 1986 with approval of the University of Bristol ethics committee. Chrna2-cre mice (Tg(Chrna2-cre)1Kldr) in which Cre recombinase is expressed under the nicotinic acetylcholine receptor α2 subunit (CHRNA2) promoter, was used to selectively target OLM interneurons[31]. To express ChR2 or tdTomato within Chrna2 expressing interneurons C57/Bl6 homozygous Ai32 mice (Gt(ROSA)26Sor^tm32(CAG-COP4*H134R/EYFP)Hze Jax Stock number: 024109) or Ai14 mice (Gt(ROSA)26Sor^tm14(CAG-tdTomato)Hze Jax Stock number: 007908) were bred with Chrna2-cre mice creating heterozygous

offspring with OLM specific expression of ChR2 or tdTomato. For in vivo experiments targeting OLM interneurons, Chrna2-cre mice were injected with Cre-specific viruses detailed below. Mice were group housed on a 12 h light/dark cycle (lights on at 8am) for electrophysiological experiments and on a reversed 12 h dark/light cycle (lights on at 8 pm) for mice undergoing behaviour experiments. Animal holding rooms were controlled with average ambient temperature of 21 °C and 45% humidity. Male and female mice were used for all experiments. Disaggregation of data for sex was not possible due to small sample sizes.

### Slice preparation
Brain slices were prepared from mice aged 4 to 24 weeks following cervical dislocation and decapitation and brains removed and dissected in ice-cold cutting solution (in mM: 205 Sucrose, 10 Glucose, 26 NaHCO$_3$, 2.5 KCl, 1.25 NaH$_2$PO$_4$, 0.5 CaCl$_2$, 5 MgSO$_4$), continually bubbled with 95% O$_2$ and 5% CO$_2$. Sagittal brain slices, 400 μm thick containing the hippocampus were prepared via a VT1200 vibratome (Leica). Brain slices were transferred to aCSF (in mM: 124 NaCl, 3 KCl, 24 NaHCO$_3$, 1.25 NaH$_2$PO$_4$ 10 Glucose, 2.5 CaCl$_2$, 1.3 MgSO$_4$), continually bubbled with 95% O$_2$ and 5% CO$_2$ and incubated at 35 °C for 30 min before being stored at room temperature for at least 30 min before use.

### Whole-cell patch clamp
Brain slices prepared from Chrna2-Cre x Ai32 animals were transferred to a submerged slice recording chamber with a constant 2.5 ml/min flow of aCSF, held at 32 °C. Slices were visualised using infra-red DIC optics on an Olympus BX-50WI microscope for LTP experiments and a SliceScope Pro 6000/Multiphoton Imaging System (Scientifica) for 2-photon imaging experiments. Patch electrodes with a resistance of 3–6 MΩ were pulled from borosilicate filamented glass capillaries (1.5 OD × 0.86 ID × 100 L mm, Harvard Apparatus) with a horizontal puller (P-97, Sutter Instrument Co., UK) and filled with internal solution.

Whole cell recordings were made with a MultiClamp 700 A amplifier (Molecular Devices, USA), filtered at 2.4 kHz and digitised at 10 kHz with a CED Power1401 data acquisition board and Signal 5.12 acquisition software (CED, Cambridge, UK). Series resistance was monitored throughout all experiments and cells that showed >40% change were discarded from subsequent analysis. Recordings were also rejected from analysis if the series resistance was greater than 30 MΩ.

### Voltage clamp LTP recordings
Cells were voltage clamped at −70 mV to obtain excitatory currents. The internal pipette solution contained (in mM) 120 KMeSO$_3$, 10 HEPES, 0.2 EGTA, 4 Mg-ATP, 0.3 Na-GTP, 8 NaCl, 10 KCl and adjusted to pH 7.4, 280–285 mOsm. No correction was made for the junction potential. Two Bipolar stimulating electrodes were placed in Stratum Radiatum (SR) and one in the SLM to evoke independent excitatory synaptic responses from the SC and temporammonic (TA) pathways[68]. No distinction was made between proximal and distal areas of SR when placing stimulating electrodes. The three independent pathways denoted: SC test pathway, TA test pathway and SC control pathway were stimulated separately at 0.05 Hz. A steady 5 min baseline of EPSC amplitudes was achieved before attempting to induce plasticity via the application of a TBS to the two test pathways. The TBS performed in current-clamp configuration consisted of a train of 10 bursts where each burst contained 5 pulses at 100 Hz with the frequency of bursts set at 5 Hz. The TBS was applied through the two test pathway stimulating electrodes either in combination with light pulses following the same pattern, to activate the ChR2-expressing OLM interneurons, or in the absence of light. Inhibitory synaptic responses during the TBS were evoked optogenetically via 2 ms square light pulse via a mounted 470 nm LED (Thorlabs, US) through a 40× objective lens.

LTP induction was performed within 10 min of whole-cell configuration to prevent plasticity washout. Following TBS, responses from all pathways were recorded for a further 25 min. Consecutive traces were averaged to produce a mean response every 30 s. The mean amplitude response of the baseline period was used to normalise the responses of each pathway. Plasticity was assessed by comparing the average ESPC amplitude during the last 5 min of the experiment between the control and test pathways.

### Ex vivo two-photon Ca$^{2+}$ imaging
For dendritic 2-photon imaging experiments, an internal solution containing (in mM:130 KMeSO$_3$, 8 NaCl, 1 MgCl$_2$, 10 HEPES, 4 MgATP, 0.3 Na$_2$GTP, 5 QX-314) was supplemented with a Ca$^{2+}$ fluorescent indicator (Fluo-5F, 200 μM) and a fluorescent dye (Alexa Fluor 594, 30 μM). Whole-cell recordings of CA1 pyramidal neurons established in voltage clamp (−70 mV) were then switched to current clamp and dye allowed to defuse into the neuron for at least 20 min. Bipolar stimulating electrodes were placed in the SR and SLM regions and used to evoke bursts of synaptic stimulation (5 pulses at 100 Hz). Secondary apical dendrites in both SR (100–200 μm from the soma) and SLM (250–300 μm from the soma) regions were imaged via a 60x objective lens (Olympus) with fluorescence excitation provided via a tuneable Ti:Sapphire pulsed laser (Newport Spectra-Physics) tuned to 810 nm. Dendrites were initially visualised in raster scanning mode and during synaptic stimulation individual dendritic branches were imaged via a line scanning mode each line scan consisting of 1000 lines per second for 1 s. Synaptic stimulation and dendritic imaging was repeated 4 times for each dendrite with an interval of 20 s with the resulting somatic voltage and dendritic fluorescent traces averaged across repeats. Images were acquired with a data acquisition board (National Instruments Corporation) using ScanImage software (version 3.8). For optogenetic activation of OLM interneurons during Ca$^{2+}$ calcium imaging (5 pulses of 470 nm light coincident with electrical stimulation) was provided by a laser-diode (Doric lenses LDFLS_473/070) attached to 105 μm diameter fibre optic cannulae (Thorlabs CFMLC21L02) positioned over the SLM layer. During optical stimulation, optogenetic light was blocked from the PMT with a mechanical optical beam shutter (SHB1T, Thorlabs), positioned above the objective lens.

### Histology and immunohistochemistry
Brains were fixed via cardiac perfusion of phosphate-buffered saline (PBS) followed by 4% Formaldehyde in PBS. Brains were removed and stored in PFA for 24 h and then transferred to 30% sucrose PBS solution for 48 h. 40 μm thick sagittal slices were then obtained via freezing microtome. Slices were first washed with PBS and then incubated in a blocking solution containing 5% donkey serum and 0.2% Triton X-100 for 60 min at room temperature. Slices were subsequently incubated at 4 °C for 24 h in blocking solution containing anti-SST (1:1000 Santa Cruz SC-7819) to stain for SOM and then incubated in blocking solution containing anti-goat Alexa Fluor 488 (1:500, Invitrogen Antibodies A-11055) for 2.5 h at room temperature. Slices were then washed with PBS and mounted on microscope slides with 1:1000 DAPI staining for visualisation, and images acquired using a widefield fluorescence microscope.

To visualise viral expression and lens implant locations, Brains were fixed via cardiac perfusion with a high calcium Phosphate-buffered saline solution (in mM: 137 NaCl, 2.7 KCl, 8.1 Na$_2$HPO$_4$, 1.47 KH$_2$PO$_4$, 0.49 MgCl*6H$_2$0, 0.9 CaCl, pH 7.4) to activate GCaMP6, followed by 4% Formaldehyde in PBS. Brains were removed and stored in PFA for 24 h before 100 um coronal brain slices were obtained via a VT1200 vibratome (Leica). Slices were washed in PBS containing 1:1000 DAPI mounted on microscope slides and imaged via a widefield fluorescence microscope.

## Viruses

For $Ca^{2+}$ imaging of OLM interneurons Chrna2-cre mice were injected with a Cre dependent GCaMP6s (AAV1.Syn.Flex-GCaMP6s-WRPE-SV40, Addgene #100845 titre: $1.6 \times 10^{13}$ GC/ml). For $Ca^{2+}$ imaging of pyramidal neurons and manipulation of OLM interneurons, Chrna2-cre mice were injected with GCaMP6f under the CAMKII promotor (AAV1-CAMKII-GCAMP6f-WRPE-SV40 Addgene #100834, titre: $1:10-2.8 \times 10^{13}$ GC/ml), along with either Cre dependent redshifted excitatory opsins ChrimsonR (AAV5-Syn-Flex-rc[ChrimsonR-tdTomato] Addgene # 62723 titre: $8.5 \times 10^{12}$ GC/ml), redshifted inhibitory opsin Jaws (AAV-CAG-FLEX-rc[Jaws-KGC-tdTomato-ER2], Addgene #84446 titre $7.9 \times 10^{12}$ GC/ml constructed by the VVF, Zurich) or the control reporter mCherry (AAV5-hsyn-DIO-mcherry, Addgene #50459 titre: $1:4-7 \times 10^{12}$ GC/mL).

## In vivo surgical procedures

C57BL/6 Chrna2-cre mice aged 5–8 weeks underwent 2 surgical procedures. First stereotaxic injections of adeno-associated viruses (AAV) were conducted under (1.5–4%) isoflurane and 0.05 mg/kg buprenorphine. Viruses were injected via a 35 G NanoFil needle (WPI) into the right hippocampus CA1 (anteriorposterior: −1.9, mediolateral: +1.5 and dorsoventral: −1.5 relative to Bregma). Each injection had a volume of 600 nl and injected at 50 nl/min. The needle was left in place for 10 min post injection to allow viruses to diffuse from the injection site before removing the needle.

Mice were left a week to recover post-viral injection surgery after which they underwent GRIN lens and baseplate implantation surgery conducted under (1.5–4%) isoflurane, 0.05 mg/kg buprenorphine and 5 mg/kg Carprofen. First, a 1.5 mm craniotomy was drilled above the viral injection site. Cortex above the hippocampus was aspirated under constant irrigation with ice-cold aCSF (in mM: 150 NaCl, 2.5 KCl, 10 HEPES, 1 $CaCl_2$, 1 $MgCl_2$, pH 7.3). Upon reaching the mediolateral white matter striations a 1 mm diameter 4 mm length integrated lens and baseplate (ProView integrated lens, Inscopix) was slowly lowered on top of the hippocampus (anteroposterior: −2.0, mediolateral: +1.5 and dorsoventral: −1.15 relative to Bregma). The lens was fixed to the skull first with a layer of super-bond (Sun Medical) followed by surrounding bone cement containing gentamicin (CMW 1, DePuy synthesis). Mice were allowed to recover for at least 3 weeks post-surgery before behavioural training.

## Linear track behaviour experiments

Water-restricted mice (0.0375 ml/g per day with a minimum 85% initial body weight) were trained to run back and forth along a 140 cm linear track with a 6 μl 10% sucrose reward delivered within a 15 cm reward zone at the track ends. Three distinct linear tracks were used each with a unique texture and distinct local cues on the track walls. Distal cues were provided by two sets of distinct curtains surrounding each track, providing track polarity. Each experiment day consisted of three sessions. For familiar experiments, three consecutive sessions on a familiar track and for novel experiments, one session on a familiar track followed by two consecutive sessions on a novel track. Each 10 min session was separated by a 3 min home cage rest within an opaque box beside the track and familiar tracks were considered familiar after completion of at least five separate consecutive sessions on the same track. Mice were handled, habituated and trained to run on the track by introducing the mice to the track for 30 min everyday for 8 days. Post hoc tracking of the animal position was achieved by tracking the position of a miniscope-mounted LED in custom MATLAB software. We then used the mouse position coordinates to calculate the mouse velocity, which was then smoothed using a moving average filter with a width of 2 s.

## In vivo $Ca^{2+}$ imaging

$Ca^{2+}$ imaging of hippocampal neurons was performed via 1-photon miniaturised microscopes (nVoke 2.0, Inscopix). The miniscope was attached to the baseplate and animals allowed to move freely along the linear track via custom counter balanced pully system. $Ca^{2+}$ imaging was acquired via 455 nm excitation wavelength LED to activate the GCaMP calcium indicator. Recordings were obtained at resolution of $1280 \times 800$ pixels at a rate of 20 frames a second (20 Hz), with a field of view of 1050 μm x 650 μm. For each animal the focal plane, LED intensity and gain imaging parameters were established prior to the first day of recording and kept constant throughout all experiments. Optogenetic stimulation was provided through the miniscope via a (620 nm) LED light. To activate the excitatory opsin ChrimsonR, A continual train of 5 ms light pulses at 20 Hz were applied at a light intensity of 20 $mW/mm^2$. For activation of inhibitory opsin Jaws a continual square pulse at light intensity 5 $mW/mm^2$ was applied. Animal behaviour tracking was achieved via webcam (C920S, Logitech) positioned above the track controlled via custom written Python programme that enabled automated reward delivery and optogenetic triggering via a COM port connection to an Arduino Uno. The Arduino relayed TTL pulses to the Inscopix DAQ board to trigger optogenetic stimulation to the miniscope and to two sperate solenoid valves to trigger reward delivery. Near real time tracking was achieved with a delay of 1 video frame (33 ms), which was accounted for via staggered optogenetic stimulation zone for each lap direction. Behavioural recordings were acquired at 30 fps and synchronised to the start and end of $Ca^{2+}$ imaging sessions via a sync LED in the behaviour video field of view.

## $Ca^{2+}$ recording image processing

$Ca^{2+}$ recordings were analysed via the Inscopix data processing software (IDPS) and Python API (version 1.9.5). Raw video files were temporally and spatially down sampled to 10 Hz and $320 \times 200$ pixels and filtered with a Gaussian blur bandpass filter (high-pass cutoff: 0.5/pixel and low-pass cutoff 0.005/pixel). Each video frame was then motion corrected to an initial reference frame. Motion correction was restricted to a region of interest around the fluorescent field of view.

To obtain cell footprints of excitatory pyramidal neurons Constrained Nonnegative Matrix Factorisation for micro-endoscopic data (CNMFe)[69] was applied via the IDPS software. Hyperparameters for the CNMFe algorithm were selected for each animal to maximise cell separation and minimise over segmentation of cell footprints and were kept consistent for each field of view. $Ca^{2+}$ transients were initially identified as $Ca^{2+}$ signals that had a rising amplitude above 9 times the median absolute deviation of the trace (MAD) or a rate of decay equal to or slower than an exponential decay with a tau of 0.2. Cell footprints with a spatial component above 1, a signal to noise ratio less than 10 and an event rate of less than 0.005 Hz were discarded from further analysis, any remaining footprints were then manually inspected to ensure they were accurately detecting neuronal signals. The resulting cell footprint traces were then deconvolved using the Online Active Set method to Infer Spikes (OASIS) algorithm[70] applied via the IDPS software. We used a single order model and resulting spikes with an amplitude below a SNR of 6 were removed from the exported spike times used for further analysis. The OASIS algorithm approximates the number and timing of 'spikes' from the $Ca^{2+}$ fluorescent traces, although this algorithm is not accurate in detecting ground truth action potential spike times[71], it acts as a more accurate approximation for temporal burst activity of pyramidal neurons compared to single $Ca^{2+}$ transient time points.

For $Ca^{2+}$ imaging of OLM interneurons, cell footprints were manually identified and the resulting raw fluorescent traces extracted from the video. A baseline for each fluorescent trace was calculated by applying a 10-s sliding window and selecting the lowest 1st percentile of fluorescence values within each window as the baseline. This baseline was then used to calculate the $\Delta F/F_0$ for each interneuron trace.

## Longitudinal registration

Neurons were matched across recording sessions using the IDPS registration algorithm, as described in ref. 72. Cell maps were aligned to the first field of view using the enhanced correlation coefficient image registration algorithm[73]. Then, cells were matched by identifying cell pairs that maximise the normalised cross correlation (NCC) between their spatial footprints. Cell matching was performed incrementally; cell pairs with the highest NCC were matched first and this process was repeated until no cell pairs with an NCC above 0.5 remain. We chose the correlation threshold of 0.5[74], as this appropriately balances false-positive and false-negative errors.

## OLM activity

To obtain the lap-by-lap OLM interneuron activity, ΔF/F0 Ca²⁺ activity of OLM interneurons was thresholded to contain only activity when the mouse was running above 7 cm/s. For each interneuron an average activity per lap was obtained, which was then averaged across neurons. Lap activities were taken from the last 10 laps of session 1 and first 8–10 laps of session 2. For average lap velocity this was taken as the average running velocity (velocity higher than 7 cm/s) for each mouse.

## Place cell identification

Spike activity inferred from the recorded Ca²⁺ transients were assigned a track coordinate based on the mouse position across one behavioural session. Due to the directionality of place cell firing fields[75] activity during left and right traversals of the track were analysed separately and activity was only included for running epochs where the mouse velocity exceeded 7 cm/s for a minimum of 10 cm. The two 15 cm reward zones at the end of the track were also excluded from the analysis The resulting neuron spike activity was binned and averaged into 40 spatial bins giving a 2.75 cm bin width. This activity was divided by the mouse's occupancy within the bin and gaussian filtered ($\sigma = 1.5$). The resulting rate maps were used to classify a neuron as a place cell by calculating the spatial information for each neuron.

$$Spatial\ information = \sum_i p_i(r_i/\bar{r}).\log_2(r_i/\bar{r}) \tag{1}$$

where $p_i$ and $r_i$ is probability and firing rate of the mouse/cell in position bin $i$ and $\bar{r}$ the mean firing rate.

This spatial information value was then compared to a distribution of spatial information values acquired by shuffling the spike times 1000 times. Statistically significant spatial coding was calculated using a z-test comparing the actual spatial information value to the bootstrapped spatial information distribution, alpha set at 0.05. For neurons that were classified as place cells for both left and right lap direction the rate map for the direction that had the lowest firing rate was discarded. All other rate maps for left and right traversals were normalised to the maximum firing bin and combined for all subsequent analysis. Place field centres were calculated as the bin with the maximal normalised activity. Place field bins are defined as the consecutive bins from the place field centre that have an activity rate equal to or exceeding 75% of the event rate in the centre bin. Place field widths are defined as the sum of the bin widths in the place field. Cross-validation of place fields was performed on the initial familiar track dataset. Fields for even and odd laps were separated and sorted by even lap activity and showed high spatial and population vector correlation (Fig. S6A, B).

## Population vector correlations

To compare the similarity of spatial representations across sessions, we calculated the population vector correlation between sessions. The population vector is defined as the activity rate for each neuron at a given positional bin. These population vectors are then correlated (using Pearson's correlation) for each positional bin, resulting in a pairwise matrix of correlation values. For comparisons between sessions, the population vector activity is defined as the activity of place cells in session 1, with the same cells' activity, irrespective of place cell status, in session 2. Registration of cells across days (and therefore different fields of view) necessarily omitted cells that were inactive on one of the days and therefore population vector correlations were not possible across days. The population correlations for optogenetic and control zones are computed by taking the diagonal elements of the correlation matrix, corresponding to each zone, and calculated from the rate maps of each animal.

## Spatial correlation calculations

To compare the similarity between a cells spatial firing a spatial correlation for each cell is calculated as the Pearson's correlation between the place cell activity in session 1 with the with the same cells' activity, irrespective of place cell status, in session 2. If the neurons activity in session 2 is zero, the spatial correlation is excluded from the analysis.

## Place cell retention

Place cell retention between sessions was calculated by taking the sorted place cells from the initial session and comparing them to the same cells in subsequent sessions. Place cells were classed as lost if the place cell in the initial session was not classified as a place cell in subsequent sessions. Place cells were considered retained if the place cell in the initial session was also a place cell in subsequent sessions. These retained place cells were then split into stable place cells if the place field centres of the two cells were within 2 spatial bins and remapped place cells if the place field centres were more than 2 spatial bins apart. When comparing place cell retention to a previous session (Fig. 4L), de novo place cells were classified as place cells in the subsequent session that were not place cells in the previous session.

## Statistical analysis

Experimental unit was defined by analysis of the source of greatest variance in the data. For ex vivo experiments and place cell and OLM interneuron in vivo measurements the experimental unit is cell and for behavioural experiments and population vector correlation analysis it is animal. Data in the text and in the figures are presented as mean ± SEM unless otherwise stated. The level of significance was assigned * if $p < 0.05$, ** if $p < 0.01$ and *** if $p < 0.001$ for statistical comparisons of all datasets. Data were processed and analysed using custom MATLAB and Python code.

## Reporting summary

Further information on research design is available in the Nature Portfolio Reporting Summary linked to this article.

# Data availability

The data generated in this study are provided in the source data file. Data are also available at https://github.com/mellor-lab/OLM_and_placecell_project and from the corresponding author Jack.Mellor@bristol.ac.uk. Source data are provided with this paper.

# Code availability

Code is available at https://github.com/mellor-lab/OLM_and_placecell_project and from the corresponding author Jack.Mellor@bristol.ac.uk.

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

## Acknowledgements

We thank Sarah Stuart for training in surgical procedures, Feng Xuan and Daniel Dombeck for advice with analysis and Klas Kullander for providing Chrna2-cre mice. We thank Claudia Clopath and David Dupret for input to conceptual development, Paul Chadderton, Peter Dayan and Daniel Dombeck for input to previous versions of the manuscript and all members of the Mellor group for discussion. This work was supported by Biotechnology and Biological Sciences Research Council (BBSRC, BB/N013956/1, BB/N019008/1), Wellcome Trust (101029/Z/13/Z, 108899/B/15/Z), Medical Research Council (MRC, MR/X010910/1) awarded to J.R.M.

## Author contributions

Conceptualisation, M.U. and J.R.M.; methodology, M.U., M.C., H.-W.Z. and E.O.; investigation, M.U., M.C., H.-W.Z. and E.O.; visualisation, M.U., M.C., H.-W.Z. and E.O.; writing—original draft, M.U. and J.R.M.; writing—review and editing, M.U., M.C., H-W. Z., E.O. and J.R.M.; funding acquisition, M.U. and J.R.M.; supervision, J.R.M.

## Competing interests

The authors declare no competing interests.
