## [Transparent peer review file · Nature Communications]

Hippocampal OLM interneurons regulate CA1 place cell plasticity and remapping

Corresponding Author: Professor Jack Mellor

Version 0:

Reviewer comments:

Reviewer #1

(Remarks to the Author)

This study highlights the role of OLM interneurons in regulating synaptic plasticity by inhibiting large dendritic calcium transients triggered by concurrent CA3 and entorhinal cortex inputs. The study also shows that these neurons shape hippocampal place cell plasticity and spatial stability in new environments. OLM interneurons. Notably, OLM activity decreases in novel settings, allowing plasticity to occur. The research also demonstrates that adjusting OLM activity influences place cell formation and stability proposing a dynamic model in which OLM interneurons create a critical window for synaptic plasticity, enabling the hippocampus to refine spatial maps based on new experiences. The experimental design is clearly outlined, the OLM imaging and simultaneous PC calcium imaging/OLM manipulation are particularly relevant technical achievements

including the crucial step of registering individual cells across sessions to ensure consistent measurement. I only have a couple of concerns that I believe should be addressed before publication:

* The number of animals used should be stated alongside the number of cells in the initial figures.

* Explain how the same neurons were imaged across sections.

* I would presume that the excitation wavelength of the calcium imaging would produce some level of excitation/inhibition of OLM cells expressing ChrimsonR/JAWs as there is some overlap of the excitation spectra of these opsins. If the authors did not test this effect, at least discuss this possibility.

(Remarks on code availability)

The authors use analysis techniques that are mostly possible to reproduce (e.g. Matlab for place cell analysis)

While I am not familiar with Inscopix data processing software, the CNMFe for analysing single photon calcium imaging is more or less standard. It is not possible

To use the Online Active Set method to Infer Spikes (OASIS) algorithm should be referenced.

I could not review the code as the github link for the code (https://github.com/mellor-lab/OLM_project_analysis) provided does not exist.

Reviewer #2

(Remarks to the Author)

The manuscript by Udakis et al. demonstrates a key role for somatostatin-expressing OLM interneurons in mediating synaptic plasticity and influencing place cell formation and stability in CA1 pyramidal neurons. This is a compelling and rigorous study, and I particularly like the integration of cellular, microcircuit, and circuit-level analyses using patch clamp physiology, 2-photon imaging, and optogenetics combined with one-photon in vivo imaging analysis. The findings here position OLM interneurons as critical gatekeepers of long-term potentiation (LTP) in CA1 via their regulation of dendritic calcium dynamics. Notably, the authors show that OLM interneuron activity is suppressed when mice explore novel environments. Using miniscope imaging in CA1 combined with optogenetic inhibition/ excitation of OLM interneurons, they

further demonstrate the role these cells play in place cell formation and stability during exposure to novel contexts. While the study is excellent overall, there are a few moderate concerns.

1) First, it is not clear why plateau potential (large calcium events) that occurred during LTP induction are not shown and quantified in figure 1. Since a major claim of this manuscript is that OLM neurons affect dendritic calcium, the area under the burst should be compared for OLM stim + TBS and TBS alone experiments, and the example traces should be shown in the manuscript.

Although the authors provide area-under-the-curve (AUC) quantification in Figure 2 (C–E), they do not show how OLM interneurons modulate the coincident activation of Schaffer collateral (SC) and temporoammonic (TA) pathway inputs during multiple bursts of TBS (LTP induction protocol). Including the quantification of AUC during the LTP burst would further strengthen the claim that OLM activity gates supralinear dendritic calcium events, thereby influencing LTP induction in CA1 pyramidal neurons.

2) In the experiments shown in figure 1 and 2, it is unclear if the authors stimulated the SC inputs in the proximal or distal parts of stratum radiatum (SR). Similarly, did the authors differentiate between the dendrites located in distal vs. proximal SR for imaging experiments shown in Figure 2. The authors should report the location of the SC stimulating electrodes (Fig. 1) and the SR dendrites imaged (Fig. 2).

3) The analyses in Fig. 3 demonstrate that the speed of movement decreases in novel environment and this reduction in speed is correlated with reduced OLM activity during novel environment exploration. The authors mention in the results section “OLM activity is positively modulated by mouse speed and negatively modulated by novelty”. As mice complete multiple laps in the novel environment, novelty would be expected to decrease, and so the decrease in mouse velocity and OLM activity would be much lower at the end of the session in comparison to the beginning of the new environment recording session. I am curious if the authors have compared these metrics between the start and end of the recording sessions.

From the plots in Fig. 3 C, it appears that the peak-to-peak variance in OLM activity is much higher in laps 1 to 10 compared to laps 10 to 20. This appears to be the case for familiar-to-novel switch recordings and also for continuous recordings in a familiar environment.

4) I'm curious if the authors assessed the mouse velocity in experiments shown in figures 4 and 5. If OLM activity increase/decrease alters the mapping of a novel space, does it also affect exploratory behavior. This can be answered by comparing the mouse velocity in opto zone and control zone for both opto stim and opto inhibition experiments in figures 4 and 5.

5) In figure 4L, both the control zone and opto zone population vector correlations seem to be increased when OLM activity is increased. If this difference is significant, it would suggest a short-term plasticity in OLM inhibitory synapses leading to a persistent effect on CA1 neuronal output. Can the authors compare and report the population vector correlations for control zone between the Novel and Novel OLM stim conditions?

6) Statistical comparisons are missing for Fig. 3G, 4G, and 5E and should be included.

(Remarks on code availability)

Reviewer #3

(Remarks to the Author)

The manuscript by Udakis et al. aimed to describe the role of OLM interneurons in regulating representations of environment by modulating the activity of hippocampal place fields. OLM neurons receive unique long range alpha-adrenergic input and synapse onto distal dendrites of CA1 pyramidal neurons and PV neurons in the hippocampus. Prior work has revealed their importance in regulating plasticity of pyramidal neurons in ex vivo preparations as well as hippocampal dependent learning during behavior. The authors sought to understand how they shape population representations in vivo using a combination of ex vivo LTP experiments, as well as measurement and manipulation of their activity in vivo during behavior. The authors demonstrate that OLM neuron activity modulates hippocampal plasticity via their influence on distal dendrite activity, and that the activity of these neurons is reduced during exploration of a novel environment. The authors show that optogenetic stimulation of these neurons drives increased stability of hippocampal place fields, and argue that this stability is driven by decreased plasticity of pyramidal neurons innervated by stimulated OLM interneurons. Overall, the manuscript seeks to describe a circuit mechanism by which the remapping of place cells in the hippocampal CA1 may be regulated. This builds on prior work showing that OLM interneurons are critical for spatial working memory and context-associated learning. Previous works have shown OLM neuron's to be necessary for spatial working memory and context-associated learning. The work presented is novel in providing a circuit mechanism by which spatial representational drift may be controlled in the CA1. There are some shortcomings with regards to rigor detailed below that I believe could be addressed.

Major Comments:

1. It appears that the activity of pyramidal neurons also decreases during exploration of a familiar environment (ie session 1 vs session 2 in 4G and indirect comparison of overall activity in session 1 of Figure 4 to session 1 of figure 5). It would be helpful to quantify differences in the activity of pyramidal neurons in novel vs familiar environments as was done for OLM neurons in Figure 3 to better understand whether the decrease in activity during exploration of novel environment is global or

- specific for OLM interneurons, particularly since the populations are reciprocally connected.
2. The authors claim that decreasing OLM neuron activity facilitates place cell remapping and increasing OLM activity promotes stability (figure 4). While the increased stability following OLM stimulation appears to be statistically validated in 4H and L it is not clear that there has been a statistical test supporting the converse claim.
 3. It is not clear to me why OLM stimulation decreases activity of pyramidal cells during session 1 but not session 2; is a single session enough to render the maze 'familiar'?
 4. The authors make the statement that OLM Stimulation decreases CA1 cell activity in familiar environments but this is not statistically tested (line 497). The authors should perform statistical validation of the data from 3G and 4E to make this claim
 5. The n used for calculating statistics is in cases not clear – for instance in 2C is the n 4 mice or 35 cells. It would be helpful to include a panel for each experiment indicating the number of cells recorded from each mouse.
 6. There are a number of cases in which the authors chose to use an uncorrected t test when the correct statistical test would appear to be an anova with posthoc correction (eg Figure 3i, 3L, 4i, 4K)
 7. There are a number of figures such as 3J in which proportions of cells are quantified but no statistical test is performed and it is unclear if the authors are highlighting difference or sameness.
 8. In figure 6, when investigating the role of OLM activation on place cell persistence over days, it is unclear how multi-day registration of recorded cells was performed. Could the authors elaborate on how they validated that cells were the same cells on day 1 and 2 of recordings?
 9. The schematic of plasticity induction in 1B could be made clearer by orienting time along the horizontal axis (some of the rows (Pre (SC), Pre (TA) and sometimes Light) occur together, whereas others happen at other times (ie. Post, and Time).
 10. It is unclear if any statistical test was performed on data from 5K, though the data seems to be the strongest support of the authors claim that OLM activity influences place cell representations in familiar environments.
 11. In figure 1A and B the combination of color scheme and font size on the schematic render it almost impossible to read. Similarly, the contrast could be increased in the red panel in 1A, and the colors of the grayscale panels changed to match the wide view, multicolored panel.
 12. A schematic of the behavioral experiment would be useful in figure 3, as would histology showing GRIN lens location (in 3A). Similarly, the color of the familiar and novel tracks could be made to match the color scheme in other figures.
 13. The microscopy pictures in figure 4C and 4E are too small to be useful. Similarly, the experimental scheme in these panels is too small to be informative. Additionally, in the experimental schema, please indicate the control zone as well as the opto zone.
 14. I find the quantification of place cells in control vs opto zone to be confusing, and moreover there is no statistical testing for instance is the 10% place cells in opto vs 24% in control different (figure 4B but also corresponding figure in 3D)?

Minor Comments:

1. In figure 1C and 1D, please clarify the time point at which the paired t-test is being run.
2. In figure 2A would replace Ca²⁺ with dendritic Ca²⁺
3. The unit of Stim intensity should be made clear in 2C-F
4. It is unclear how mice are being habituated to the familiar environment.
5. Fonts are unreadably small in a number of panels in Figure 4.
6. In line 294, cells' needs a possessive apostrophe.
7. The proportion bars in 4C, 4D, 4E, and 4F do not have a color key, and it is unclear what the large grey portions of the bars are intended to represent.
8. OLM stimulation seems to clearly decrease firing rate in 4G (as well as 5E). It would be useful to compare activity during this period to baseline.
9. In 4I it would be helpful to include mean +/- error bars. It is unclear if the authors corrected for multiple comparisons.
10. In 4K, please clarify on the figure that each axis on the confusion matrices corresponds to session 1 or session 2.
11. The title of figure 5 is confusing given the conclusions drawn in the text (line(s) 398-399: "...the role of OLM interneurons in modulating place cell formation is restricted to conditions in which the mouse is experiencing a novel environment.>").
12. In 5J the retention of place cells in control zone should be compared to the retention in opto-zone. Similarly, the data in the left hand panel appears to be presented as a time series (why is the gap between s0 and S1 different than the gap between S1 and S2) but appears to have no additional information not include in the right hand panel (and no statistical test) and the y axis is different (starts at 20) which is confusing
13. The n is not clear for proportions displayed in figure 5k, is this the average or is this pooled cells?
14. Similarly, the multi-color scheme in 5k is confusing, particularly given that the curved font is so small, I wonder if multiple shades of purple and green are needed on the bar graphs.
15. Font sizes in figure 6 are again very small.
16. As above, in 6A and 6B, please indicate the control zone in the behavioral schema.
17. Please indicate the number of cells and the number of mice used in figure 6 in the figure legend.
18. In lines 824 and 826, the word "mouse" is misspelled ("mous").

(Remarks on code availability)

There is code that is available on authors github but not in the indicated folder.
It has a readme that explains how to run code and dependencies.

Reviewer #4

(Remarks to the Author)

General remarks

In the present study, the authors address the question how synaptic plasticity of hippocampal place cells is regulated during

the exploration of a novel environment. Using patch clamp recordings and parallel optogenetics in brain slices, they find that OLM interneurons inhibit dendritic calcium influx and suppress the potentiation of excitatory synaptic inputs from CA3 at the Schaffer collaterals that coincide with entorhinal inputs via the temporoammonic pathway. Using single-photon microendoscope Ca²⁺ imaging in mice freely navigating on a linear track they reveal that the activity of OLM interneurons is down-regulated in novel environments. Optogenetic activation of OLM interneurons increases the stability of spatial representations in novel environments and makes CA1 place cells less likely to remap during subsequent recording sessions. They conclude that their results “demonstrate a critical role for dendrite targeted inhibition for control of synaptic plasticity that underpins the remapping of neuronal representations necessary for adapting to unfamiliar environments”.

The role of interneurons in controlling the formation of spatial representations in novel environments has received much interest in recent years. While a decreased activity of CA1 SOM interneurons in novel environments has recently been reported (e.g. PMIDs 31591960, 33022227, 38267409), their functional role in controlling spatial representations has only been assessed in the dentate gyrus (PMID 38267409) but not in CA1. Therefore the present paper makes an important contribution to our understanding how interneuron subpopulations control behaviorally driven plasticity to form novel spatial representations. It provides a mechanism for a long-standing hypothesis first proposed by Wilson & McNaughton more than 30 years ago: “The suppression of inhibitory interneurons thus might facilitate the synaptic modification necessary to encode new spatial information.” (PMID 8351520).

The authors use carefully designed experiments to convincingly demonstrate that OLM interneurons are important in gating synaptic plasticity *in vitro* and are modulated by novelty and speed *in vivo*. Their experimental design aligns well with the research question, their techniques are state-of-the-art and it is impressive to see the confluence of *in vitro* and *in vivo* data. The experiments were conducted rigorously with proper controls. However, we have major concerns especially regarding the identification and analysis of place cells and place fields that need to be addressed.

Major points

1. OLM photostimulation reduces event rates in the stimulation zone in both novel (Fig. 4G) and familiar (Fig. 5E) environments by about 30%. Place cell identification is then based on a z-test comparing spatial information in the data with a synthetic data set obtained by shuffling spikes 1000 times.
 - a. It is not clear which sessions are used for place cell identification. Did they use all or just one session (e.g. S1 in Fig. 4 and 5A-H, and S0 in Fig. 5I)?
 - b. If the authors included the session with optogenetic stimulation, how do they account for the reduced firing rate in the opto zone when they shuffle spikes? An unbiased shuffling procedure that distributes spikes throughout the track would result in similar firing rates in the opto zone and outside of the opto zone for the synthetic data set. However, the lower firing rates of place cells with fields in the opto zone would affect their spatial information (because the mean firing rate is computed across the whole track) and they would pass the place cell criterion less easily than place cells with fields outside of the opto zone. This could result in a selection of place cells with particularly high firing rates among all place cells that are present in the opto zone. These neurons may represent a subset of particularly stable neurons, which could explain that spatial correlations in the opto zone are higher for the OLM stim population compared to the control population, i.e. one of the main findings of this manuscript could be explained by such an analysis bias. Fig S3 is consistent with this interpretation, as it shows that the absolute number of place cells is reduced in the opto zone although it has the same extent as the control zone, which is consistent with a stricter place cell criterion in the opto zone.
 - c. While the authors discuss the concern that “OLM activation specifically inhibits place cells that are more unstable leaving a population of stable place cells” (l. 526-528), they do not provide convincing arguments relieving our concern. The population of neurons they analysed (e.g. in 4J and 5K) is selected based on their spatial activity expressed in a session where the activity of those neurons was suppressed in a spatially confined way. This approach may therefore lead to a bias in the selection of neurons that are termed “place cells”, i.e. not only may OLM activation preferentially inhibit place cells that are more unstable (as discussed by the authors) but also the analysis approach might discard unstable place cells during OLM activation.
 - d. What is the significance level used to determine a place cell? ($\alpha = 0.05$)

Minor points

1. a. How does photo-inhibition and activation of OLM cells affect overall activity rates of principal neurons? Figs. 4G and 5E show data along the track but a summary for activity rates of principal neurons during photostimulation is missing.
- b. How did the authors assess the efficiency of photo-stimulating OLM cells? This is particularly relevant for the inhibition experiments as there is no apparent increase in pyramidal cell activity upon photo-inhibition (Fig. 5E) despite a higher OLM interneuron activity (Fig. 3C-E) in familiar environments.
2. l. 496-497: The authors state that optogenetic reduction of OLM interneuron activity in a novel environment increased remapping, referring to Fig. 4 and S3. However, there is no statistical test supporting this claim. Analogous to major point 1, the results could be biased due to a less strict place cell inclusion criterion, if some neurons were more active in the opto stim zone after OLM interneuron inhibition.
3. a. Overall activity levels of place cells in novel environments seem to be higher in session 1 vs 2 in novel (Fig. 4G), but not in familiar environments (Fig. 5E). The authors should quantify this difference and discuss the implications for the effect of OLM-interneuron activity on CA1 activity levels.

- b. How do overall activity levels between session 1 vs 2 change in all principal cells (not just place cells)?
4. How do the authors account for multiple place fields when analysing place field shifts of neurons (Fig. 4I)? Snake plots (4B-F) reveal that place cells often have more than one place field.
 5. Fig. 4K: different color codes for stimulation vs. inhibition sessions make it difficult to compare both correlation matrices.
 6. The pie charts shown in Figs. 4J and 5K are confusing. Why are there cells losing spatial tuning in Fig. 4J but not in Fig. 5K, and why are there “De novo place cells” in Fig. 5K but not in Fig. 4J? The authors could use S2 vs S1 to identify “de novo” place cells in Fig. 4J as well.
 7. The authors show activity rates during environmental transitions in Figure 3C for OLM interneurons. What do these data look like for principal neurons?
 8. Have the authors assessed stability within a session? Is intra-session stability consistent with across-session stability?
 9. l. 237: What is the time interval between sessions 1 and 2 in Figures 4 and 5?
 10. l. 303-306: the authors claim bidirectional control of the proportion of unstable place cells referencing Fig. S3 C,D. However, they do not provide direct evidence that they significantly changed the proportion of unstable cells following inhibition. (see also major point 1 for biased selection of analysed place cells)
 11. l. 363-365: although the authors provide indirect evidence supporting a successful activation of OLM neurons via ChrimsonR (decrease of place cell activity in the opto zone in Fig. 4G), they lack evidence of a successful inhibition. This was neither done experimentally at the cellular level of OLM neurons, nor quantified indirectly from their action on the population of pyramidal cells.
 12. l. 386-388: “Interestingly, OLM interneuron stimulation during session 1 led to a more drastic reduction in place cell retention to a comparable degree of place cell retention observed in the subsequent session 2. (Figure 5I,J; $61.91 \pm 3.88\%$ vs $48.34 \pm 5.86\%$ control zone, $50.96 \pm 6.84\%$ vs $54.81 \pm 8.71\%$ opto zone).” As the activity of place cells with firing in the opto zone might potentially be reduced, it is not surprising that “place cell retention” will show a “more drastic” reduction under these conditions (see also major issue #1); besides, is this statement backed up by statistics (i.e. is S1 control zone significantly different from S1 Opto zone in Fig. 5J bar graphs)?
 13. The title of Fig. 5 is misleading. The authors only provide evidence that optogenetically increased OLM activity affects spatial representations in familiar environments in panel J. In panel 5J-K they do not show the same quantifications for the inhibition of OLM interneurons. This is particularly interesting as the authors have previously shown that OLM interneuron activity is higher in the familiar environment. Thus, if they play an active role in sustaining stable place cell activity, their silencing should result in even lower place cell retention. Also, without showing that inhibition of their physiological activity does alter spatial representations, the claim made in the title of the figure is not substantiated by their findings.
 14. In lines 508-525 the authors discuss that their findings support potentiation of inhibitory synapses of SOM interneurons onto pyramidal neurons. They base this on the finding that pairing of temporoammonic input with plateau potentials in CA1 pyramidal cells resulted in LTD of the TA input. From this the authors conclude that plateau potentials cause a simultaneous increase in inhibition and decrease in excitation for synaptic transmission to distal dendrites. However, they do not provide evidence for such inhibitory synaptic plasticity since they do not record IPSCs. It is difficult to follow their argumentation towards the mechanisms by which LTP, LTD, OLM-interneuron activity, as well as inhibitory synaptic plasticity interact in stabilizing spatial firing in familiar vs. novel environments.
 15. l. 440-441: the drift occurring for novel environments across days has been documented in Vaidya et al. 2025 (previously preprint, now published in Nature Neuroscience)
 16. l. 546: which evidence suggests that non-OLM α 2 OLM interneurons respond to novelty and control dendritic excitability in the same way as OLM α 2 interneurons? The effects and rate changes reported here concern only Chrna2-cre positive (OLM α 2) interneurons.
 17. l. 110 “Having established selective control of OLM interneurons” - at this point in the manuscript, the authors only established selective expression in OLM interneurons, without yet addressing the efficacy of manipulating their activity.
 18. Fig. 2A-B: DFA is not a widely used abbreviation for dF/F amplitudes. It should be spelled out in the legend or alternatively “dF/F” should be used.
 19. l. 193-194: it has previously been shown that BTSP is driving synaptic plasticity leading to place cell formation and occurring more frequently in novel environments (e.g. Priestley et al. Neuron 2022 and Madar et al. Nature Neuroscience 2025)
 20. Fig. 3 B, the scale and unit are missing for the x axis
 21. The numbers reported should be reduced to significant digits throughout the manuscript. (e.g. p values in l. 172 or spatial correlation and place field size in l. 291-292).
 22. l.37, “Location specific place cell activity allows quantifiable measurement of the neuronal representation of an environment and when a new environment is encountered”, what do the authors mean with “quantifiable measurement of the neuronal representation”? That the neuronal representation can be experimentally quantified? If so, this statement seems oddly out of context at this point of the introduction and is neither motivated by the previous nor by the following sentence.
 23. l.784 “imagine processing, please correct.
 24. l.824 and elsewhere: Change “mous” to “mouse”
 25. l.825 “activity was only including”, please correct.

(Remarks on code availability)

The URL https://github.com/mellor-lab/OLM_project_analysis returns code 404

Reviewer #5

(Remarks to the Author)

(Remarks on code availability)

Reviewer #6

(Remarks to the Author)

(Remarks on code availability)

The link provided does not connect to the project repository.

Version 1:

Reviewer comments:

Reviewer #1

(Remarks to the Author)

The authors did a satisfactory job answering my concerns. It is my opinion that the article should be published.

(Remarks on code availability)

My impression in analysing the code is that the authors are proficient in the methods they describe

Reviewer #2

(Remarks to the Author)

The authors have thoroughly addressed my comments, and the revised manuscript is notably improved.

(Remarks on code availability)

Reviewer #3

(Remarks to the Author)

The authors have largely answered my concerns. I would like to reiterate that certain panels in some figures are too small to be read and understood. I understand that they may be legible at 100% zoom on some computers and in some programs, but this paper will presumably be read by many people on different monitors and with different processing programs. Or, readers may elect to print the work

and may not be able to zoom. Additionally, including high resolution figures is useful, but there are some panels with fonts so small that when I zoom in, I still cannot read the labels (4C, 4E).

(Remarks on code availability)

Code appears to be functional

Reviewer #4

(Remarks to the Author)

In the revised version of their manuscript, the authors have addressed most of the issues raised in the previous round of reviews.

Some minor issues still remain (numbering refers to the issues raised in the previous round):

Minor point 1a, "How does photo-inhibition and activation of OLM cells affect overall activity rates of principal neurons? Figs. 4G and 5E show data along the track but a summary for activity rates of principal neurons during photostimulation is missing."

The authors write in their reply: "This is an important point also raised by reviewers 2 and 3. We now include reference to the statistical tests in the legends for figures 4 and 5."

However, we failed to find a direct comparison of activity rates with and without OLM activity modulation in figures 4 or 5.

Minor point 1b, "How did the authors assess the efficiency of photo-stimulating OLM cells? This is particularly relevant for the inhibition experiments as there is no apparent increase in pyramidal cell activity upon photo-inhibition (Fig. 5E) despite a higher OLM interneuron activity (Fig. 3C-E) in familiar environments."

The authors write in their reply that they "did still observe a non-significant increase in pyramidal cell firing rates with Jaws activation in both novel and familiar environments". Where is this shown?

Minor point 7, "The authors show activity rates during environmental transitions in Figure 3C for OLM interneurons. What do these data look like for principal neurons?"

The authors write in their reply: "This point is answered in response to reviewer 3 point 1 and reviewer 4 minor point 3a above." The temporal dynamics of neuronal activity rates upon transitions to a novel environment are not shown for principal cells, only for interneurons (Figure 3C). It would be interesting to compare these dynamics between principal neurons and interneurons. For example, does the activity rate increase upon the first lap in the novel environment and then remain constant in principal neurons? We agree with reviewer 3 major point 1 that "It would be helpful to quantify differences in the activity of pyramidal neurons in novel vs familiar environments as was done for OLM neurons in Figure 3", and this includes Figure 3C.

(Remarks on code availability)

Reviewer #5

(Remarks to the Author)

(Remarks on code availability)

Reviewer #6

(Remarks to the Author)

(Remarks on code availability)

Response to reviewers' comments for manuscript Udakis et al 2025. Comments in blue, responses in black.

Reviewer #1 (Remarks to the Author):

This study highlights the role of OLM interneurons in regulating synaptic plasticity by inhibiting large dendritic calcium transients triggered by concurrent CA3 and entorhinal cortex inputs. The study also shows that these neurons shape hippocampal place cell plasticity and spatial stability in new environments. OLM interneurons. Notably, OLM activity decreases in novel settings, allowing plasticity to occur. The research also demonstrates that adjusting OLM activity influences place cell formation and stability proposing a dynamic model in which OLM interneurons create a critical window for synaptic plasticity, enabling the hippocampus to refine spatial maps based on new experiences. The experimental design is clearly outlined, the OLM imaging and simultaneous PC calcium imaging/OLM manipulation are particularly relevant technical achievements including the crucial step of registering individual cells across sessions to ensure consistent measurement. I only have a couple of concerns that I believe should be addressed before publication:

* The number of animals used should be stated alongside the number of cells in the initial figures.

Animal numbers are now included in the figure legends for Figures 1 and 2. It was rare to record from more than one cell per animal so the numbers for animals and cells are similar.

* Explain how the same neurons were imaged across sections.

We apologise for this omission. A description of the approach is now included in the methods (1827) and is very similar to that developed in Bollimunta et. al (PMID: 34133921).

* I would presume that the excitation wavelength of the calcium imaging would produce some level of excitation/inhibition of OLM cells expressing ChrimsonR/JAWs as there is some overlap of the excitation spectra of these opsins. If the authors did not test this effect, at least discuss this possibility.

The overlap in excitation spectra means there is potentially continual excitation of Chrimson/Jaws with 455nm light used for Ca²⁺ imaging. To minimise this crosstalk we ensured that GCaMP6 expression was high allowing us to use a blue light intensity of <20% max. Studies have shown that this intensity produces minimal excitation of optogenetic actuators such as Chrimson and Jaws (eg. Stamatakis et al 2018 PMID: 30087590). Furthermore, our experiments explicitly control for this by using within experiment comparisons between opto-stim and control zones of the track. We now state this explicitly in the results section 1271.

Reviewer #1 (Remarks on code availability):

The authors use analysis techniques that are mostly possible to reproduce (e.g. Matlab for place cell analysis)

While I am not familiar with Inscopix data processing software, the CNMFe for analysing single photon calcium imaging is more or less standard. It is not possible to use the Online Active Set method to Infer Spikes (OASIS) algorithm should be referenced.

The OASIS algorithm from Friedrich et al 2017 (PMID: 28291787) is now referenced at the correct place in the methods (I813).

I could not review the code as the github link for the code (https://github.com/mellor-lab/OLM_project_analysis) provided does not exist.

Many apologies for this error. The code was available on our Github site (as found by reviewer 3) but we gave the wrong link. This has now been corrected and the code can be accessed at https://github.com/mellor-lab/OLM_and_placecell_project.

Reviewer #2 (Remarks to the Author):

The manuscript by Udakis et al. demonstrates a key role for somatostatin-expressing OLM interneurons in mediating synaptic plasticity and influencing place cell formation and stability in CA1 pyramidal neurons. This is a compelling and rigorous study, and I particularly like the integration of cellular, microcircuit, and circuit-level analyses using patch clamp physiology, 2-photon imaging, and optogenetics combined with one-photon in vivo imaging analysis. The findings here position OLM interneurons as critical gatekeepers of long-term potentiation (LTP) in CA1 via their regulation of dendritic calcium dynamics. Notably, the authors show that OLM interneuron activity is suppressed when mice explore novel environments. Using miniscope imaging in CA1 combined with optogenetic inhibition/ excitation of OLM interneurons, they further demonstrate the role these cells play in place cell formation and stability during exposure to novel contexts. While the study is excellent overall, there are a few moderate concerns.

1) First, it is not clear why plateau potential (large calcium events) that occurred during LTP induction are not shown and quantified in figure 1. Since a major claim of this manuscript is that OLM neurons affect dendritic calcium, the area under the burst should be compared for OLM stim + TBS and TBS alone experiments, and the example traces should be shown in the manuscript.

Although the authors provide area-under-the-curve (AUC) quantification in Figure 2 (C–E), they do not show how OLM interneurons modulate the coincident activation of Schaffer collateral (SC) and temporoammonic (TA) pathway inputs during multiple bursts of TBS (LTP induction protocol). Including the quantification of AUC during the LTP burst would further strengthen the claim that OLM activity gates supralinear dendritic calcium events, thereby influencing LTP induction in CA1 pyramidal neurons.

We now show these data in Figure S1 (and describe at I162) that confirm our findings in Figure 2. The AUC for depolarisation during the entire LTP induction protocol where coincident stimulation is given to both SC and TA pathways is not different between control and OLM stimulation conditions. These data from LTP experiments are from separate

experiments so the comparison is unpaired. The result is similar to the within experiment comparisons made in Figure 2 showing no difference in depolarisation AUC when coincident SC and TA inputs are stimulated to generate plateau potentials in the presence or absence of OLM stimulation. The critical effect of OLM stimulation is to decrease dendritic calcium as shown in Figure 2.

2) In the experiments shown in figure 1 and 2, it is unclear if the authors stimulated the SC inputs in the proximal or distal parts of stratum radiatum (SR). Similarly, did the authors differentiate between the dendrites located in distal vs. proximal SR for imaging experiments shown in Figure 2. The authors should report the location of the SC stimulating electrodes (Fig. 1) and the SR dendrites imaged (Fig. 2).

We used bipolar stimulating electrodes with tips 100µm apart. Therefore, the effective area of stimulation for electrodes placed in SR encompasses most of the layer and we cannot clearly distinguish proximal or distal regions of SR that would be activated. This is now described in the methods (I663). SR dendrites were chosen as secondary apical dendrites in a range of 100-200µm from the soma. This is now stated in the methods (I690).

3) The analyses in Fig. 3 demonstrate that the speed of movement decreases in novel environment and this reduction in speed is correlated with reduced OLM activity during novel environment exploration. The authors mention in the results section “OLM activity is positively modulated by mouse speed and negatively modulated by novelty”. As mice complete multiple laps in the novel environment, novelty would be expected to decrease, and so the decrease in mouse velocity and OLM activity would be much lower at the end of the session in comparison to the beginning of the new environment recording session. I am curious if the authors have compared these metrics between the start and end of the recording sessions.

This is an interesting question. We find that the average velocity across the whole track, which is a reasonable measure of exploratory behaviour, drops in a novel environment and then stays constant for >10 laps of the novel session. We now show these data in Figure S2A of the revised manuscript. This indicates that the novelty of the new environment persists for at least 1 session and mirrors the reduced OLM activity which also decreases and remains at the decreased level for the remainder of the session (Figure 3).

From the plots in Fig. 3 C, it appears that the peak-to-peak variance in OLM activity is much higher in laps 1 to 10 compared to laps 10 to 20. This appears to be the case for familiar-to-novel switch recordings and also for continuous recordings in a familiar environment.

Further analysis of the variance in these data indicates that this is not the case. These plots are the average OLM interneuron activity rate on each lap of the track. Whilst the lap-to-lap variability of the average data may appear higher during laps 1-10 the actual variance on each individual lap (standard error of the mean for each point) is fairly constant when comparing laps 1-10 and laps 11-20. The lower variance for individual laps observed in a novel environment is as expected for a lower average.

4) I'm curious if the authors assessed the mouse velocity in experiments shown in figures 4

and 5. If OLM activity increase/decrease alters the mapping of a novel space, does it also affect exploratory behavior. This can be answered by comparing the mouse velocity in opto zone and control zone for both opto stim and opto inhibition experiments in figures 4 and 5.

Analysis of mouse velocity revealed no difference between opto and control zones for the data shown in Figures 4 and 5. This agrees with our observation of behaviour during these experiments where no overt changes could be seen in light stim regions and is perhaps as expected for unilateral stimulation in just a small number of OLM interneurons stimulated in a targeted region of the hippocampus. This result is now shown in Figure S2B and referred to in the text (I287).

5) In figure 4L, both the control zone and opto zone population vector correlations seem to be increased when OLM activity is increased. If this difference is significant, it would suggest a short-term plasticity in OLM inhibitory synapses leading to a persistent effect on CA1 neuronal output. Can the authors compare and report the population vector correlations for control zone between the Novel and Novel OLM stim conditions?

The PVCs in the control zone appear higher in the Novel OLM stim condition than Novel condition but when an ANOVA is performed on the control zone data across these 2 conditions there is no statistical difference ($p = 0.22$). Even if there was, this would not be a reasonable comparison to make because there are several different novel tracks that we use in the study and the ones used for Novel and Novel OLM stim conditions are different. Therefore, the comparison across conditions is not appropriate since different tracks may have intrinsically different stability of representations and there may also be external factors that alter stability of representations from one session to another. That is why we focus our analysis on within experiment comparisons between opto stim and control zones of the track.

6) Statistical comparisons are missing for Fig. 3G, 4G, and 5E and should be included.

We thank the reviewer for pointing out this omission. The statistical comparison for Fig 3G is quantified as correlation coefficients and shown in Fig 3H. P values for paired t-tests of opto stim effect on pyramidal neuron event rates in Fig 4G and 5E are now given in the figure legends.

Reviewer #3 (Remarks to the Author):

The manuscript by Udakis et al. aimed to describe the role of OLM interneurons in regulating representations of environment by modulating the activity of hippocampal place fields. OLM neurons receive unique long range alpha-adrenergic input and synapse onto distal dendrites of CA1 pyramidal neurons and PV neurons in the hippocampus. Prior work has revealed their importance in regulating plasticity of pyramidal neurons in ex vivo preparations as well as hippocampal dependent learning during behavior. The authors sought to understand how they shape population representations in vivo using a combination of ex vivo LTP experiments, as well as measurement and manipulation of their activity in vivo during behavior. The authors demonstrate that OLM neuron activity modulates hippocampal

plasticity via their influence on distal dendrite activity, and that the activity of these neurons is reduced during exploration of a novel environment. The authors show that optogenetic stimulation of these neurons drives increased stability of hippocampal place fields, and argue that this stability is driven by decreased plasticity of pyramidal neurons innervated by stimulated OLM interneurons. Overall, the manuscript seeks to describe a circuit mechanism by which the remapping of place cells in the hippocampal CA1 may be regulated. This builds on prior work showing that OLM interneurons are critical for spatial working memory and context-associated learning

Previous works have shown OLM neuron's to be necessary for spatial working memory and context-associated learning. The work presented is novel in providing a circuit mechanism by which spatial representational drift may be controlled in the CA1. There are some shortcomings with regards to rigor detailed below that I believe could be addressed.

Major Comments:

1. It appears that the activity of pyramidal neurons also decreases during exploration of a familiar environment (ie session 1 vs session 2 in 4G and indirect comparison of overall activity in session 1 of Figure 4 to session 1 of figure 5). It would be helpful to quantify differences in the activity of pyramidal neurons in novel vs familiar environments as was done for OLM neurons in Figure 3 to better understand whether the decrease in activity during exploration of novel environment is global or specific for OLM interneurons, particularly since the populations are reciprocally connected.

The increase in CA1 pyramidal cell activity in response to novelty is a well characterised phenomenon (see for example Priestley et al 2022 PMID: 35447088, Larkin et al 2014 PMID: 24596296). This enhanced activity is generalised across the place cell population and gradually subsides as the environment becomes more familiar. On average pyramidal cell firing rates were $20 \pm 5\%$ ($p < 0.01$, $n = 13$ mice) higher in novel environments vs familiar environments. These data are now reported in the text (l278). Therefore, in a novel environment OLM activity decreases and pyramidal cell activity increases.

2. The authors claim that decreasing OLM neuron activity facilitates place cell remapping and increasing OLM activity promotes stability (figure 4). While the increased stability following OLM stimulation appears to be statistically validated in 4H and L it is not clear that there has been a statistical test supporting the converse claim.

Our data predict that reducing OLM activity will promote synaptic plasticity and by inference place cell remapping, as occurs in a novel environment, but the reviewer is correct that we have not tested this directly. We have adjusted the sentence at l389 to reflect this.

3. It is not clear to me why OLM stimulation decreases activity of pyramidal cells during session 1 but not session 2; is a single session enough to render the maze 'familiar?'

This may be a misunderstanding. We only stimulate OLM interneurons in session 1 and not in session 2. Therefore, we would not expect a decrease in pyramidal cell activity in session 2.

The track is not “familiar” after 1 session (although it is less novel) but the familiarity of the track does not determine the reduction in pyramidal cell activity caused by OLM interneuron stimulation in the opto zone (cf Figure 4G vs 5E).

4. The authors make the statement that OLM Stimulation decreases CA1 cell activity in familiar environments but this is not statistically tested (line 497). The authors should perform statistical validation of the data from 3G and 4E to make this claim

This is a good point also raised by rev2 point 6 and answered above. The statistical tests are now reported in the manuscript figure legends (related to Figures 4G and 5E – we assume this was the reviewer’s intended figures).

5. The n used for calculating statistics is in cases not clear – for instance in 2C is the n 4 mice or 35 cells. It would be helpful to include a panel for each experiment indicating the number of cells recorded from each mouse.

The cell and animal numbers are now reported for all experiments in Figures 1 and 2 at the appropriate places in the figure legends. The n used for statistics in these figures is cells. For *in vivo* data in Figures 3-6 we report cell and animal numbers and clarify in the figure legend the n used for statistical analysis. See also accompanying spreadsheet with all data and statistical tests.

6. There are a number of cases in which the authors chose to use an uncorrected t test when the correct statistical test would appear to be an anova with posthoc correction (eg Figure 3i, 3L, 4i, 4K)

Our approach to statistical testing is described in the methods where experimental unit was defined by analysis of the source of greatest variance in the data. For *ex vivo* experiments and place cell and OLM interneuron *in vivo* measurements the experimental unit is cell and for behavioural experiments and population vector correlation analysis it is animal. It is not entirely clear to us which results the reviewer is referring to (there are no figures 3i and 3L in the manuscript). However, we can explain our rationale for using t-tests rather than ANOVA for data shown in Figures 4-6. The experiments are designed to specifically test the differences between opto and control zones within experiment. Between experiments the tracks are different and therefore not immediately comparable. This makes the tests a direct comparison and so we use a t-test.

7. There are a number of figures such as 3J in which proportions of cells are quantified but no statistical test is performed and it is unclear if the authors are highlighting difference or sameness.

We assume this refers to the pie charts shown in Figures 4J and 5K. These are percentages of cells from all animals. The result shown in Figure 4J is not significant and we apologise for not including this result. It is now included in the text (l310). The result shown in Figure 5K (now supplementary Figure S5) is also not significant and these data were reported in the text (l415).

8. In figure 6, when investigating the role of OLM activation on place cell persistence over days, it is unclear how multi-day registration of recorded cells was performed. Could the authors elaborate on how they validated that cells were the same cells on day 1 and 2 of recordings?

We apologise for this omission. A description of the approach is now included in the methods (1827) and is very similar to that developed in Bollimunta et. al (PMID: 34133921).

9. The schematic of plasticity induction in 1B could be made clearer by orienting time along the horizontal axis (some of the rows (Pre (SC), Pre (TA) and sometimes Light) occur together, whereas others happen at other times (ie. Post, and Time).

The schematic has now been simplified. Time is on the horizontal axis and the SC, TA and light input timings for the LTP induction protocol are now clearly signified.

10. It is unclear if any statistical test was performed on data from 5K, though the data seems to be the strongest support of the authors claim that OLM activity influences place cell representations in familiar environments.

“We found that the percentage of *de novo* place cells (place cells in session 1 but not place cells in session 0), was not significantly changed in the optogenetic stimulation zone compared to the control zone ($18.03 \pm 5.39\%$ vs $23.31 \pm 7.23\%$, opto vs control zone, $p > 0.05$) suggesting that the role of OLM interneurons in modulating place cell formation is restricted to conditions in which the mouse is experiencing a novel environment (Figure 5K).” This is how the text currently reads to describe the statistical test performed on these data.

Similar to rev4 minor point 13 and rev 3 minor point 11 we have now simplified the narrative and therefore title of Figure 5 where the main finding is a lack of effect of OLM manipulation on place cell stability. In line with this change we have now moved Figure 5I-K into supplementary Figure S5.

11. In figure 1A and B the combination of color scheme and font size on the schematic render it almost impossible to read. Similarly, the contrast could be increased in the red panel in 1A, and the colors of the grayscale panels changed to match the wide view, multicolored panel.

We appreciate the reviewer’s comments on this. When making the figures we tried various colour options and have chosen text and label colours that have the best contrast as possible on each respective image. For the grey scale panels we wanted the viewer to compare the expression between panels directly, visually this is not possible with different colours due to human perception of different hues. As for font size we appreciate that font sizes are small but text sizes any larger would impede the image and with digital publication and the ability to zoom into high resolution figures we feel this is justified. Based on this we would prefer to keep the figure as it is.

12. A schematic of the behavioral experiment would be useful in figure 3, as would histology

showing GRIN lens location (in 3A). Similarly, the color of the familiar and novel tracks could be made to match the color scheme in other figures.

We are not sure what schematic the reviewer is intending. In Figure 3C we show a schematic of the familiar and novel tracks used in the experiment which is the only behavioural manipulation.

We agree it would be nice to show histology of the GRIN lens locations. However, these are large diameter lenses that are implanted for months and their extraction is almost impossible to achieve without significant damage to the underlying tissue. Therefore, unfortunately we have not recovered any useful images of lens implant location. Instead, we show histology demonstrating selective expression of GCaMP6, Chrimson and Jaws in OLM interneurons and GCaMP6 expression in pyramidal cells and example miniscope fields of view indicating correct location of the lenses.

There is no colour scheme for the schematics for familiar and novel tracks across figures 3-6. We feel it is more important to have colour schemes representing track zones and OLM stim vs OLM inhibition.

13. The microscopy pictures in figure 4C and 4E are too small to be useful. Similarly, the experimental scheme in these panels is too small to be informative. Additionally, in the experimental schema, please indicate the control zone as well as the opto zone.

Figure 4 is a multi-panel figure which means that some panels are smaller than we would ideally like. We have tried out a number of configurations and chose the current format to prioritise the most important data. We find that the images in Figure 4C and 4E are ok and convey the key messages at 100% zoom. We have taken care to include the images at high resolution so that readers may zoom in to greater magnification if they wish without losing quality. This is in line with other publications in this journal.

The control and opto zones of the track are indicated in the schematic in Figure 4A.

14. I find the quantification of place cells in control vs opto zone to be confusing, and moreover there is no statistical testing for instance is the 10% place cells in opto vs 24% in control different (figure 4B but also corresponding figure in 3D)?

In Figures 4C,D,E,F and 5A,B,C,D the % of place cells in the opto and control zones is simply a % of the total number of place cells on the whole track. This is indicative and we have not performed any statistical tests on these numbers. The statistical tests to show effect of opto stimulation on place cell firing rates are more informative and we now include the results of those tests in the figure legends. These results broadly agree with the %s of place cells in each zone.

Minor Comments:

1. In figure 1C and 1D, please clarify the time point at which the paired t-test is being run.

This information is given in the methods section I679 “Plasticity was assessed by comparing the average ESPC amplitude during the last 5 min of the experiment between the control and test pathways.”

2. In figure 2A would replace Ca²⁺ with dendritic Ca²⁺

Thank you, this is a more accurate description and it is now replaced.

3. The unit of Stim intensity should be made clear in 2C-F

There is no unit of stim intensity but they are kept consistent for control and OLM stim conditions and are now designated as arbitrary units (AU).

4. It is unclear how mice are being habituated to the familiar environment.

Tracks were deemed to be familiar after mice had run at least five consecutive 10 minute sessions on the track. This is described in the methods (I767). Habituation during behavioural training was achieved by introducing the mice to the track for 30 minutes every day for 8 days. Due to their natural instincts to explore a new environment the mice quickly learnt to run from end to end to receive the reward.

5. Fonts are unreadably small in a number of panels in Figure 4.

We realise they are small but as described in the response to reviewer 3 major point 13 above we are able to read all the font sizes and have included figures at high resolution for easy magnification if necessary.

6. In line 294, cells' needs a possessive apostrophe.

Thank you, now corrected.

7. The proportion bars in 4C, 4D, 4E, and 4F do not have a color key, and it is unclear what the large grey portions of the bars are intended to represent.

The colour key is between panels E and H and is consistent for the whole figure so we do not feel it is necessary to replicate. The grey zones indicate regions of the track at either end not in the opto or control zones and this is now specified in the legend.

8. OLM stimulation seems to clearly decrease firing rate in 4G (as well as 5E). It would be useful to compare activity during this period to baseline.

This is now done. Please see response to reviewer 2 comment 6 and reviewer 3 comment 4 above.

9. In 4I it would be helpful to include mean +/- error bars. It is unclear if the authors corrected for multiple comparisons.

We show the distribution histograms instead of mean +/- error bars for variety and because they are more informative in this instance so we prefer this depiction. The experiments are designed to have within experiment comparisons between control and opto zones with each condition being independent so multiple comparisons are not necessary. Also see response to rev3 major comment 6 above.

10. In 4K, please clarify on the figure that each axis on the confusion matrices corresponds to session 1 or session 2.

This is included in the figure legend.

11. The title of figure 5 is confusing given the conclusions drawn in the text (line(s) 398-399: "...the role of OLM interneurons in modulating place cell formation is restricted to conditions in which the mouse is experiencing a novel environment.").

We originally wanted to capture the observation that manipulation of OLM interneurons acutely alters place cell firing rates and therefore retention of place cells in familiar environments. However, on reflection we agree that the title was misleading and we have now changed it to "Stability of spatial representations in familiar environments is unaltered by OLM activity". A similar point is made by rev4 minor point 13.

12. In 5J the retention of place cells in control zone should be compared to the retention in opto-zone. Similarly, the data in the left hand panel appears to be presented as a time series (why is the gap between s0 and S1 different than the gap between S1 and S2) but appears to have no additional information not include in the right hand panel (and no statistical test) and the y axis is different (starts at 20) which is confusing

We have now compared retention rates between opto and control zones and find no significant difference. This is now indicated on the figure and reflects the major observation that OLM manipulation did not change place cell stability in familiar environments. We have also moved these analyses to supplementary Figure S5 since on reflection we feel that this analysis is tangential to the overall narrative of Figure 5 (related to the previous point). A similar point is made by rev4 minor point 12.

We have now corrected the axes in the left hand plot of Figure 5J. This plot shows the evolution of place cell retention across sessions whereas the bar plots on the right show the quantification of these data.

13. The n is not clear for proportions displayed in figure 5k, is this the average or is this pooled cells?

It is an average of the percentages for each animal and so the n is number of animals (5 in this case). This is now stated in the figure legend.

14. Similarly, the multi-color scheme in 5k is confusing, particularly given that the curved font is so small, I wonder if multiple shades of purple and green are needed on the bar graphs.

These shades concur for bar graphs and pie charts which we feel helps interpretation of the data and we prefer to retain this scheme.

15. Font sizes in figure 6 are again very small.

We agree they are small, but to us they are legible.

16. As above, in 6A and 6B, please indicate the control zone in the behavioral schema.

This is indicated in the schematic in Figure 4A and we do not feel it needs replicating in Figures 5 and 6.

17. Please indicate the number of cells and the number of mice used in figure 6 in the figure legend.

Thank you for pointing out this omission. The animal numbers are the same as in Figure 4 and these details are now included. Cell numbers are shown on the heat map plots.

18. In lines 824 and 826, the word "mouse" is misspelled ("mous").

Corrected.

Reviewer #3 (Remarks on code availability):

There is code that is available on authors github but not in the indicated folder. It has a readme that explains how to run code and dependencies.

Many apologies for this error. This has now been corrected and the code can be accessed at https://github.com/mellor-lab/OLM_and_placecell_project.

Reviewer #4 (Remarks to the Author):

General remarks

In the present study, the authors address the question how synaptic plasticity of hippocampal place cells is regulated during the exploration of a novel environment. Using patch clamp recordings and parallel optogenetics in brain slices, they find that OLM interneurons inhibit dendritic calcium influx and suppress the potentiation of excitatory synaptic inputs from CA3 at the Schaffer collaterals that coincide with entorhinal inputs via the temporoammonic pathway. Using single-photon microendoscope Ca²⁺ imaging in mice freely navigating on a linear track they reveal that the activity of OLM interneurons is down-regulated in novel environments. Optogenetic activation of OLM interneurons increases the stability of spatial representations in novel environments and makes CA1 place cells less likely to remap during subsequent recording sessions. They conclude that their results

“demonstrate a critical role for dendrite targeted inhibition for control of synaptic plasticity that underpins the remapping of neuronal representations necessary for adapting to unfamiliar environments”.

The role of interneurons in controlling the formation of spatial representations in novel environments has received much interest in recent years. While a decreased activity of CA1 SOM interneurons in novel environments has recently been reported (e.g. PMIDs 31591960, 33022227, 38267409), their functional role in controlling spatial representations has only been assessed in the dentate gyrus (PMID 38267409) but not in CA1. Therefore the present paper makes an important contribution to our understanding how interneuron subpopulations control behaviorally driven plasticity to form novel spatial representations. It provides a mechanism for a long-standing hypothesis first proposed by Wilson & McNaughton more than 30 years ago: “The suppression of inhibitory interneurons thus might facilitate the synaptic modification necessary to encode new spatial information.” (PMID 8351520).

The authors use carefully designed experiments to convincingly demonstrate that OLM interneurons are important in gating synaptic plasticity *in vitro* and are modulated by novelty and speed *in vivo*. Their experimental design aligns well with the research question, their techniques are state-of-the-art and it is impressive to see the confluence of *in vitro* and *in vivo* data. The experiments were conducted rigorously with proper controls. However, we have major concerns especially regarding the identification and analysis of place cells and place fields that need to be addressed.

Major points

1. OLM photostimulation reduces event rates in the stimulation zone in both novel (Fig. 4G) and familiar (Fig. 5E) environments by about 30%. Place cell identification is then based on a z-test comparing spatial information in the data with a synthetic data set obtained by shuffling spikes 1000 times.

a. It is not clear which sessions are used for place cell identification. Did they use all or just one session (e.g. S1 in Fig. 4 and 5A-H, and S0 in Fig. 5I)?

It is just one session that is used for place cell identification. This is now clarified in the methods (l846).

b. If the authors included the session with optogenetic stimulation, how do they account for the reduced firing rate in the opto zone when they shuffle spikes? An unbiased shuffling procedure that distributes spikes throughout the track would result in similar firing rates in the opto zone and outside of the opto zone for the synthetic data set. However, the lower firing rates of place cells with fields in the opto zone would affect their spatial information (because the mean firing rate is computed across the whole track) and they would pass the place cell criterion less easily than place cells with fields outside of the opto zone. This could result in a selection of place cells with particularly high firing rates among all place cells that are present in the opto zone. These neurons may represent a subset of particularly stable neurons, which could explain that spatial correlations in the opto zone are higher for the

OLM stim population compared to the control population, i.e. one of the main findings of this manuscript could be explained by such an analysis bias. Fig S3 is consistent with this interpretation, as it shows that the absolute number of place cells is reduced in the opto zone although it has the same extent as the control zone, which is consistent with a stricter place cell criterion in the opto zone.

The reviewer raises an important point, but we do not believe it is significant for the following 3 reasons:

1. If we look at the data from familiar environments where place cell event rates are depressed by a similar amount by OLM interneuron stimulation (Figure 5), we do not see any stabilisation of the place cells in the stimulation zone.
2. If we were selecting high firing place cells as proposed we would expect these cells to have increased spatial information, but this is not the case (Figure S5).
3. We have now performed an additional analysis using the control mCherry dataset in novel environments (Figure S4) where we artificially reduced the spike rate in the opto zone by 30% to mimic the effect of OLM interneuron stimulation by randomly removing spikes in the opto zone. This produced a similar reduction in the number of detected place cells when we ran these new datasets through our analysis pipeline. However, we did not see any stabilisation of the remaining place cells in subsequent sessions. Therefore, we conclude that the reduction in spike rate cannot explain the observed stabilisation of remaining place cells in the OLM stim track region. We now include this new analysis in Figure S4 and describe it at I538.

c. While the authors discuss the concern that “OLM activation specifically inhibits place cells that are more unstable leaving a population of stable place cells” (l. 526-528), they do not provide convincing arguments relieving our concern. The population of neurons they analysed (e.g. in 4J and 5K) is selected based on their spatial activity expressed in a session where the activity of those neurons was suppressed in a spatially confined way. This approach may therefore lead to a bias in the selection of neurons that are termed “place cells”, i.e. not only may OLM activation preferentially inhibit place cells that are more unstable (as discussed by the authors) but also the analysis approach might discard unstable place cells during OLM activation.

This is related to the previous point and we refer to the response above.

d. What is the significance level used to determine a place cell? (alpha = 0.05?)

Yes alpha = 0.05. This is now stated in the methods (I862).

Minor points

1. a. How does photo-inhibition and activation of OLM cells affect overall activity rates of principal neurons? Figs. 4G and 5E show data along the track but a summary for activity rates of principal neurons during photostimulation is missing.

This is an important point also raised by reviewers 2 and 3. We now include reference to the statistical tests in the legends for figures 4 and 5.

b. How did the authors assess the efficiency of photo-stimulating OLM cells? This is particularly relevant for the inhibition experiments as there is no apparent increase in pyramidal cell activity upon photo-inhibition (Fig. 5E) despite a higher OLM interneuron activity (Fig. 3C-E) in familiar environments.

We have not performed the gold standard experiment of measuring OLM interneuron activity rates while stimulating or inhibiting them *in vivo*. This would require dual expression of GCaMP and opsin in OLM interneurons and therefore a completely new cohort of mice. Given the widespread characterisation of these tools and our data showing good expression of opsins in OLM interneurons combined with the changes in pyramidal cell firing rates we observed with OLM stimulation we felt there was sufficient evidence that we were successfully activating the opsins in each case. The effect of OLM inhibition on place cell firing rates are less clear but we did still observe a non-significant increase in pyramidal cell firing rates with Jaws activation in both novel and familiar environments. It is not surprising that the changes are smaller than for ChrimsonR activation since the OLM interneurons are predicted to only affect pyramidal cell firing rates in response to entorhinal input. Our current model proposes that this input is only significant during new learning events and therefore OLM inhibition effect on pyramidal firing rates will be hard to detect in our behavioural paradigm. Interestingly, the condition where this might become most apparent is focusing specifically on unstable and stable place cells in recently novel environments. This is precisely where we see the strongest effects of OLM inhibition which increases the proportion of unstable place cells in direct contrast to OLM stimulation (Figure S6C,D).

2. I. 496-497: The authors state that optogenetic reduction of OLM interneuron activity in a novel environment increased remapping, referring to Fig. 4 and S3. However, there is no statistical test supporting this claim. Analogous to major point 1, the results could be biased due to a less strict place cell inclusion criterion, if some neurons were more active in the opto stim zone after OLM interneuron inhibition.

In the discussion we state that reducing OLM interneuron activity in a novel environment increased the proportion of unstable place cells. Statistical tests for this claim are reported in Figure S6D related to the data shown in Figure 4. These tests show a strong trend which does not reach statistical significance with the low sample size. Given the opposite significant effect of OLM excitation shown in Figure S6C we feel the statements made in the results (I318) and discussion (I502) are not unreasonable given this analysis.

Analogous to point 1, we do not believe the results are biased by the place cell inclusion criteria for the reasons given above.

3. a. Overall activity levels of place cells in novel environments seem to be higher in session 1 vs 2 in novel (Fig. 4G), but not in familiar environments (Fig. 5E). The authors should quantify this difference and discuss the implications for the effect of OLM-interneuron activity on CA1 activity levels.

This is similar to reviewer 3 point 1. The increase in CA1 pyramidal cell activity in response to novelty is a well characterised phenomenon (see for example Priestley et al 2022 PMID: 35447088, Larkin et al 2014 PMID: 24596296). This enhanced activity is generalised across the place cell population and gradually subsides as the environment becomes more familiar. On average pyramidal cell firing rates increased by $20 \pm 5\%$ ($p < 0.01$, $n = 13$ mice) in a novel environment. These data are now reported in the text (I278). Therefore, in a novel environment OLM activity decreases and place cell activity increases.

b. How do overall activity levels between session 1 vs 2 change in all principal cells (not just place cells)?

The activity levels shown in Figure 4G are for all pyramidal cells.

4. How do the authors account for multiple place fields when analysing place field shifts of neurons (Fig. 4I)? Snake plots (4B-F) reveal that place cells often have more than one place field.

If there are multiple place fields in one cell our analysis takes the highest firing place field as the principal place field and analysis is performed as if the cell had just one place field. There is a low incidence of place cells with 2 place fields in the same track direction and if a cell has a field in the right and the left running direction then the direction with the lowest event rate field is excluded. This is stated in the methods (I845).

5. Fig. 4K: different color codes for stimulation vs. inhibition sessions make it difficult to compare both correlation matrices.

This is perhaps true but it keeps consistency across the figure and therefore assists with comparisons. We also quantify the PVCs in Figure 4L. Therefore, we prefer to keep the colour scheme as it is.

6. The pie charts shown in Figs. 4J and 5K are confusing. Why are there cells losing spatial tuning in Fig. 4J but not in Fig. 5K, and why are there “De novo place cells” in Fig. 5K but not in Fig. 4J? The authors could use S2 vs S1 to identify “de novo” place cells in Fig. 4J as well.

The pie charts shown in Figures 4J and 5K show two different things.

In 4J we are looking forward from session 1 to session 2 to analyse what happens to place cells in the opto session 1. Many of the place cells from session 1 are no longer place cells in session 2, some stay the same (stable) and others move (remapped). Since this is a novel environment and we are not analysing the cells in session 0 technically all these cells are *de novo* place cells although we do not describe them as such and this might be confusing.

In 5K we are looking backwards at what the place cells in session 1 looked like in session 0. Some of them are new place cells that were not present in session 0 and these we term *de novo*, some stay in the same place (stable) and others move (remapped). After consideration, we now feel that the impact of the data shown in Figure 5I-K is not high so we have moved this to Figure S5.

7. The authors show activity rates during environmental transitions in Figure 3C for OLM interneurons. What do these data look like for principal neurons?

This point is answered in response to reviewer 3 point 1 and reviewer 4 minor point 3a above.

8. Have the authors assessed stability within a session? Is intra-session stability consistent with across-session stability?

We have not assessed stability within session beyond performing the cross validation shown in Figure S6A,B. These data show that the place cell representations are reasonably stable within a session but this is not quantified across time within the session. Whilst within session stability is an interesting question it is not directly relevant to the questions we ask in this manuscript. Here we seek to understand the impact of stimulating or inhibiting OLM interneurons in one session on the stability of place cells in subsequent sessions. Also, possibly because of the intervention where we take animals off the track between sessions the major adaptation of place cells occurs between sessions rather than within session. There are published papers that specifically address the question of within session stability of place cells, eg. Dong et al 2021 PMID: 34016996, Cohen et al 2017 PMID: 28742496.

9. I. 237: What is the time interval between sessions 1 and 2 in Figures 4 and 5?

It is 3 minutes as stated in the methods I765. We now restate this for clarity in the results I284.

10. I. 303-306: the authors claim bidirectional control of the proportion of unstable place cells referencing Fig. S3 C,D. However, they do not provide direct evidence that they significantly changed the proportion of unstable cells following inhibition. (see also major point 1 for biased selection of analysed place cells)

This is similar to the comment of reviewer 4 minor comment 2 above and is answered there.

As discussed above for major comment 1, our evidence shows that biased selection of place cells is not an issue for our results and conclusions.

11. I. 363-365: although the authors provide indirect evidence supporting a successful activation of OLM neurons via ChrimsonR (decrease of place cell activity in the opto zone in Fig. 4G), they lack evidence of a successful inhibition. This was neither done experimentally at the cellular level of OLM neurons, nor quantified indirectly from their action on the population of pyramidal cells.

This is similar to the comment of reviewer 4 minor comment 1b above and is answered there.

12. I. 386-388: "Interestingly, OLM interneuron stimulation during session 1 led to a more drastic reduction in place cell retention to a comparable degree of place cell retention

observed in the subsequent session 2. (Figure 5I,J; $61.91 \pm 3.88\%$ vs $48.34 \pm 5.86\%$ control zone, $50.96 \pm 6.84\%$ vs $54.81 \pm 8.71\%$ opto zone).” As the activity of place cells with firing in the opto zone might potentially be reduced, it is not surprising that “place cell retention” will show a “more drastic” reduction under these conditions (see also major issue #1); besides, is this statement backed up by statistics (i.e. is S1 control zone significantly different from S1 Opto zone in Fig. 5J bar graphs)?

This is a good point and we agree that the activity of place cells will be reduced in the opto zone of session 1 which explains the reduced retention in session 1. We state this in the text immediately after reporting these analyses (I404).

There is no significant difference in place cell retention between opto and control zones in session 1, what we have done is a more nuanced analysis for the rate at which place cell drift occurs across sessions measured by place cell retention. However, on reflection we feel that this analysis is tangential to the overall narrative of Figure 5 and that our statements here were too strong. Therefore, we have now adjusted the statements (I405) and moved Figure 5I-K to supplementary Figure S5. Also related to rev3 point 12.

13. The title of Fig. 5 is misleading. The authors only provide evidence that optogenetically increased OLM activity affects spatial representations in familiar environments in panel J. In panel 5J-K they do not show the same quantifications for the inhibition of OLM interneurons. This is particularly interesting as the authors have previously shown that OLM interneuron activity is higher in the familiar environment. Thus, if they play an active role in sustaining stable place cell activity, their silencing should result in even lower place cell retention. Also, without showing that inhibition of their physiological activity does alter spatial representations, the claim made in the title of the figure is not substantiated by their findings.

A similar point is made by rev3 point 11. We originally wanted to capture the observation that manipulation of OLM interneurons acutely alters place cell firing rates and therefore retention of place cells in familiar environments. However, on reflection we agree that the title was misleading and we have now changed it to “Stability of spatial representations in familiar environments is unaltered by OLM activity”.

14. In lines 508-525 the authors discuss that their findings support potentiation of inhibitory synapses of SOM interneurons onto pyramidal neurons. They base this on the finding that pairing of temporoammonic input with plateau potentials in CA1 pyramidal cells resulted in LTD of the TA input. From this the authors conclude that plateau potentials cause a simultaneous increase in inhibition and decrease in excitation for synaptic transmission to distal dendrites. However, they do not provide evidence for such inhibitory synaptic plasticity since they do not record IPSCs. It is difficult to follow their argumentation towards the mechanisms by which LTP, LTD, OLM-interneuron activity, as well as inhibitory synaptic plasticity interact in stabilizing spatial firing in familiar vs. novel environments.

This appears to be a misunderstanding. The potentiation of SOM interneuron synapses onto pyramidal cells is shown in our previous publication Udakis et al 2020 (PMID: 32879322) referenced at this point in the discussion. Further modelling in this paper predicted that inhibitory plasticity enhances place cell stability which is relevant to our results in the current

manuscript. In the discussion we combine this result with our observation in the current manuscript that excitatory TA synapses undergo LTD when OLM neurons are active during plateau potentials. We speculate that since these two forms of plasticity act to decrease excitatory-inhibitory balance in the distal dendrites they will combine to further stabilise place cells and therefore link the two findings together.

15. I. 440-441: the drift occurring for novel environments across days has been documented in Vaidya et al. 2025 (previously preprint, now published in Nature Neuroscience)

Thank you for highlighting this newly published work. We now cite this at the relevant location in the text (I448).

16. I. 546: which evidence suggests that non-OLM α 2 OLM interneurons respond to novelty and control dendritic excitability in the same way as OLM α 2 interneurons? The effects and rate changes reported here concern only Chrna2-cre positive (OLM α 2) interneurons.

We apologise for omission of citations to support this statement. The evidence for this comes from observations of OLM interneurons dissected by genetic expression profile which show similar morphologies between Chrna2 expressing and non-expressing OLM neurons (Chamberland et al 2024 PMID: 38640347) and also observations of all SST expressing OLM interneurons during exposure to novelty (Hainmueller et al 2024 PMID: 38267409; Arriaga and Han 2019 PMID: 31591960; Geiller et al 2020 PMID: 33022227). These citations are now included at this location in the discussion (I558).

17. I. 110 “Having established selective control of OLM interneurons” - at this point in the manuscript, the authors only established selective expression in OLM interneurons, without yet addressing the efficacy of manipulating their activity.

Good point and we have now reworded this to “Using this approach to selectively control OLM interneurons...”

18. Fig. 2A-B: DFA is not a widely used abbreviation for dF/F amplitudes. It should be spelled out in the legend or alternatively “dF/F” should be used.

We apologise for the confusion in terminology. The calculation is the difference in Fluo5 fluorescence from baseline divided by the Alexa594 fluorescence and we now term this dF/A consistently in the figure, legends and text.

19. I. 193-194: it has previously been shown that BTSP is driving synaptic plasticity leading to place cell formation and occurring more frequently in novel environments (e.g. Priestley et al. Neuron 2022 and Madar et al. Nature Neuroscience 2025)

These citations have now been added at this location in the text (I202).

20. Fig. 3 B, the scale and unit are missing for the x axis

Thank you. This has now been corrected.

21. The numbers reported should be reduced to significant digits throughout the manuscript. (e.g. p values in l. 172 or spatial correlation and place field size in l. 291-292).

Corrected.

22. l.37, "Location specific place cell activity allows quantifiable measurement of the neuronal representation of an environment and when a new environment is encountered", what do the authors mean with "quantifiable measurement of the neuronal representation"? That the neuronal representation can be experimentally quantified? If so, this statement seems oddly out of context at this point of the introduction and is neither motivated by the previous nor by the following sentence.

We have now reworded the first and second sentences of the introduction and hope that they are now better linked together. By quantifiable measurement of the neuronal representation we mean that the neuronal representation of an environment can be experimentally quantified.

23. l.784 "imagine processing, please correct.

24. l.824 and elsewhere: Change "mous" to "mouse"

25. l.825 "activity was only including", please correct.

All corrected.

Reviewer #4 (Remarks on code availability):

The URL https://github.com/mellor-lab/OLM_project_analysis returns code 404

Many apologies for this error. The code was available on our Github site (as found by reviewer 3) but we gave the wrong link. This has now been corrected and the code can be accessed at https://github.com/mellor-lab/OLM_and_placecell_project.

Reviewer #5 (Remarks to the Author):

Reviewer #6 (Remarks to the Author):

I co-reviewed this manuscript with one of the reviewers who provided the listed reports. This is part of the Nature Communications initiative to facilitate training in peer review and to

provide appropriate recognition for Early Career Researchers who co-review manuscripts.

Reviewer #6 (Remarks on code availability):

The link provided does not connect to the project repository.

Response to reviewers' comments for manuscript Udakis et al 2025. Comments in blue, responses in black.

Reviewer #3:

The authors have largely answered my concerns. I would like to reiterate that certain panels in some figures are too small to be read and understood. I understand that they may be legible at 100% zoom on some computers and in some programs, but this paper will presumably be read by many people on different monitors and with different processing programs. Or, readers may elect to print the work and may not be able to zoom. Additionally, including high resolution figures is useful, but there are some panels with fonts so small that when I zoom in, I still cannot read the labels (4C, 4E).

We have checked the version submitted to the reviewers and find the pdf conversion process on the online submission platform has degraded the resolution. We believe this will not be a factor in a published version and the original high resolution figures can be read without a problem. However, we agree that some of the text in Figure 4C,E is on the small side and we have now increased the font size and contrast to make these more legible. We hope these changes mitigate the reviewer's concerns on this issue.

Reviewer #4:

In the revised version of their manuscript, the authors have addressed most of the issues raised in the previous round of reviews.

Some minor issues still remain (numbering refers to the issues raised in the previous round):

Minor point 1a, "How does photo-inhibition and activation of OLM cells affect overall activity rates of principal neurons? Figs. 4G and 5E show data along the track but a summary for activity rates of principal neurons during photostimulation is missing."

The authors write in their reply: "This is an important point also raised by reviewers 2 and 3. We now include reference to the statistical tests in the legends for figures 4 and 5."

However, we failed to find a direct comparison of activity rates with and without OLM activity modulation in figures 4 or 5.

The direct statistical comparison between opto stim and control zones of the track (which corresponds to with and without OLM activity modulation) is described with p values in the Figure legend for Fig 4G and Fig 5E. The data are shown in the corresponding figure panels for the activity rates along the track. We believe this is the best way to present the data since the important comparison is location dependent and this enables visualisation of the activity rates across the whole track. Therefore, we prefer to leave the data presentation as it currently is.

Minor point 1b, "How did the authors assess the efficiency of photo-stimulating OLM cells? This is particularly relevant for the inhibition experiments as there is no apparent increase in

pyramidal cell activity upon photo-inhibition (Fig. 5E) despite a higher OLM interneuron activity (Fig. 3C-E) in familiar environments.”

The authors write in their reply that they “did still observe a non-significant increase in pyramidal cell firing rates with Jaws activation in both novel and familiar environments”. Where is this shown?

This is a similar point to the previous comment. The direct statistical comparison for the efficiency of OLM photo-inhibition is described in the Figure legends for Figures 4G and 5E and shows non-significant increases in pyramidal cell activity ($p=0.2$ for novel and $p=0.6$ for familiar environments). The data are presented as activity rates along the track in the corresponding figure panels and indicate a small increase in the opto stim zone in comparison to the control zone. For similar reasons to those given above we prefer to represent the data in this manner.

Minor point 7, “The authors show activity rates during environmental transitions in Figure 3C for OLM interneurons. What do these data look like for principal neurons?”

The authors write in their reply: “This point is answered in response to reviewer 3 point 1 and reviewer 4 minor point 3a above.” The temporal dynamics of neuronal activity rates upon transitions to a novel environment are not shown for principal cells, only for interneurons (Figure 3C). It would be interesting to compare these dynamics between principal neurons and interneurons. For example, does the activity rate increase upon the first lap in the novel environment and then remain constant in principal neurons? We agree with reviewer 3 major point 1 that “It would be helpful to quantify differences in the activity of pyramidal neurons in novel vs familiar environments as was done for OLM neurons in Figure 3”, and this includes Figure 3C.

We apologise for misunderstanding exactly what was intended in the previous round of review. We now include the temporal evolution of pyramidal cell activity rates on transition from familiar to novel environments in Figure S2C. This shows an initial increase in activity on the first lap that then recedes in subsequent laps whilst remaining higher than in the familiar environment. This is a different profile to the OLM interneuron activity rates which reduce and remain stable for the duration of the first novel environment session (Figure 3C). This illustrates that pyramidal cell activity rates in a novel environment are not solely driven by OLM activity. This is expected since many studies have demonstrated that multiple interneuron subtypes change activity rates in novel environments as described in the introduction.